# Decoding early stress signaling waves in living plants using nanosensor multiplexing

Mervin Chun-Yi Ang [1,4], Jolly Madathiparambil Saju[2,4], Thomas K. Porter [3], Sayyid Mohaideen[1], Sreelatha Sarangapani[2], Duc Thinh Khong[1], Song Wang[1], Jianqiao Cui[3], Suh In Loh[1], Gajendra Pratap Singh[1], Nam-Hai Chua[1,2], Michael S. Strano [1,3] ✉ & Rajani Sarojam [1,2] ✉

Increased exposure to environmental stresses due to climate change have adversely affected plant growth and productivity. Upon stress, plants activate a signaling cascade, involving multiple molecules like $H_2O_2$, and plant hormones such as salicylic acid (SA) leading to resistance or stress adaptation. However, the temporal ordering and composition of the resulting cascade remains largely unknown. In this study we developed a nanosensor for SA and multiplexed it with $H_2O_2$ nanosensor for simultaneous monitoring of stress-induced $H_2O_2$ and SA signals when *Brassica rapa subsp. Chinensis* (Pak choi) plants were subjected to distinct stress treatments, namely light, heat, pathogen stress and mechanical wounding. Nanosensors reported distinct dynamics and temporal wave characteristics of $H_2O_2$ and SA generation for each stress. Based on these temporal insights, we have formulated a biochemical kinetic model that suggests the early $H_2O_2$ waveform encodes information specific to each stress type. These results demonstrate that sensor multiplexing can reveal stress signaling mechanisms in plants, aiding in developing climate-resilient crops and pre-symptomatic stress diagnoses.

Plants are continually subjected to various environmental stresses, which negatively affect their growth, dramatically reducing crop yields worldwide[1,2]. The different stressors can be broadly classified as abiotic, or those brought about by adverse environmental conditions such as drought, sub-optimal light, heat stress, nutrient deficiency, and biotic stresses caused by pathogens and wounding by pests[3,4]. The frequency and intensity of these stresses are predicted to increase due to climate change reducing crop productivity and further exacerbating the critical food security situation[2]. To increase the resilience of crops to environmental stress, discerning the information content of early plant stress responses is enabling. Technologies that enable rapid detection of stress-induced biochemical changes in crops will help farmers enact timely interventions to preserve yield, and scientist to elucidate the interplay between pathways fostering the development of climate-resilient crops[5].

Plants activate complex signaling pathways upon stress perception, which later results in changes in cellular transcriptome to generate a customized physiological and metabolic response to a particular stress[6,7]. Rapid generation of reactive oxygen species (ROS) is one of the earliest signaling events that occurs in plants in response to both biotic and abiotic stresses. ROS play crucial roles in stress sensing, activation of various stress response networks, and the establishment of defense mechanisms and resilience. Recent developments of genetically encoded ROS sensors and dyes show that different stresses generate distinct stress-specific redox signatures, which along with other signals, potentially trigger stress-specific response pathways[8,9]. Stress-induced ROS can trigger the production of ROS in

[1]Disruptive & Sustainable Technologies for Agricultural Precision IRG, Singapore-MIT Alliance for Research and Technology, 1 CREATE Way, #03-06/07/08 Research Wing, Singapore 138602, Singapore. [2]Temasek Life Sciences Laboratory Limited, 1 Research Link National University of Singapore, Singapore 117604, Singapore. [3]Department of Chemical Engineering, Massachusetts Institute of Technology, 77 Massachusetts Avenue, Cambridge, MA 02139, USA. [4]These authors contributed equally: Mervin Chun-Yi Ang, Jolly Madathiparambil Saju. ✉e-mail: strano@mit.edu; rajanis@tll.org.sg

neighboring cells initiating a rapid systemic signaling ROS leading to activation of defense mechanisms[10]. Among the ROS, $H_2O_2$ has the longest chemical lifetime and is an important redox molecule involved in early plant stress signaling due to its specific chemical and physical properties and stability within the cells[8,9,11].

Plant stress also affects the steady-state levels of different hormones and studies have shown that $H_2O_2$ and hormone signaling pathways interact extensively with each other to orchestrate an appropriate stress response[11,12]. SA is a multifaceted plant hormone involved in regulating many aspects of plant growth, development, and response to stresses[13]. SA's most important and well-studied function is in mediating plant responses upon pathogen infection. Apart from inducing local defense responses upon pathogen invasion, SA is also responsible for the establishment of systemic acquired resistance (SAR) at non-infected distal parts of the plants[14–17]. SAR induction activates broad-spectrum immunity against pathogens and prevents the further spread of infections[18–20]. In addition, it primes the plants towards mounting a rapid and robust defense response to future pathogens. Research has shown that SA is also involved in mediating plant responses to major abiotic stresses like extreme temperature, drought, salinity, toxic metals, UV, and osmotic stress[21–23]. The exogenous application of SA is known to enhance stress tolerance of plants to many abiotic stresses[24–28]. Because of this role, we hypothesized that an examination of $H_2O_2$ and SA dynamics after stress stimulus may reveal distinct signatures related to the type of stress.

Studies have revealed the extensive interplay between ROS and SA during defense responses to biotic and abiotic stresses. SA is proposed to act as both pro-oxidant and antioxidant under different stress conditions, regulating ROS homeostasis in plants[11,29]. It has been shown that ROS signals can act upstream or downstream to SA signaling under stress, although the underlying mechanisms and the sequence of signaling events remains largely unknown[12,30]. Technologies that enable non-destructive real-time detection of initial biochemical signals involved in early stress response are still lacking. We note that such technology can also help early diagnosis of stresses in plants[5]. Detection of stress prior to the appearance of visual symptoms in plants provides an opportunity to intervene and take remedial actions to reduce yield loss. Hence, new and innovative technologies which can serve as an early warning system of plant stress are required. Current methods of stress detection are disruptive and involve lengthy laboratory-based tests[31,32]. Alternative sensing strategies such as chlorophyll fluorescence spectroscopy[33–35] and hyperspectral imaging[36,37] are being developed but they detect plant stress largely on the basis of metabolic changes that occurs subsequently in a plant after the initial stress perception and signaling.

In this study, we developed a single-walled carbon nanotube (SWNT) based optical nanosensor for real-time detection of SA in planta and applied it in tandem with a previously developed $H_2O_2$ nanosensor[38], to elucidate the interconnection of ROS and SA pathways in response to different environmental stimuli and stresses (Fig. 1a). The $H_2O_2$ nanosensor was developed based on SWNTs wrapped with single-stranded $(GT)_{15}$ DNA oligomer and was utilized as an optical probe for real-time monitoring of endogenous $H_2O_2$ in plants when subjected to various biotic and abiotic stress conditions[38]. SWNTs are highly photo-stable and fluoresce in the near-infrared (nIR) region away from the chlorophyll auto-fluorescence region[39]. When non-covalently bound to single-stranded DNA oligomer, $(GT)_{15}$, a corona phase is formed around the SWNT conferring specific binding ability to $H_2O_2$. This sensing strategy is known as corona phase molecular recognition (CoPhMoRe)[40]. By introducing CoPhMoRe nanosensors into living plants, nanobionic plants with sensing capabilities are successfully engineered[41–45]. The success of the CoPhMoRe strategy is now expanded in our development of a highly selective plant nanobionic

sensor for SA through a distinct and unique process of design, synthesis, and testing.

Recently, SA detection in plant tissue samples have been reported using sensors such as SA antibodies[46,47], titanium dioxide nanoparticle-based colorimetric assay[48], and electrochemical sensors[49–52]. However, they are limited respectively by the lack of spatial and temporal resolution, limited sensitivity, and need for electrode insertion causing plant tissue damage. A genetically modified strain of *Acinetobacter sp. ADP1* containing the luciferase gene was also developed as an in vivo SA biosensor, which produces bioluminescence in response to SA and methylsalicylate[53,54]. The biosensor however was demonstrated solely in model tobacco plants and was able to measure changes in SA during infection only in the apoplast of leaf cells. Our SA nanosensor was validated in planta using transgenic *A. thaliana* plants and later used to probe the spatiotemporal dynamics of SA production upon bacterial infection in non-model pak choi plants. Further, concurrent detection of both SA and $H_2O_2$ was performed by multiplexing both nanosensors in the same leaf together with a common reference sensor under different stress conditions namely light stress, heat stress, pathogen stress, and mechanical wounding. For the first time, distinct temporal patterns of local $H_2O_2$ and SA generation were observed for each specific stress within hours of stress treatment. Our data shows that sensor multiplexing can be a transformative strategy that can provide novel scientific insights on the interplay of different plant signaling pathways in counteracting various types of stresses.

## Results
### Synthesis and characterization of cationic polymer SWNT wrappings
To enable electrostatic interactions with anionic plant hormones, 4 cationic fluorene-based co-polymers (S1 to S4) were synthesized as SWNT wrappings (Supplementary Fig. 1a). Previous CoPhMoRe screening using fluorene-based co-polymers have yielded a selective turn-on nanosensor for the synthetic auxin compound, 2,4-dichlorophenoxyacetic acid (2,4-D)[44]. The selective 2,4-D nanosensor consists of a cationic fluorene monomer copolymerized with 1,3-phenyl monomer. In place of the 1,3-phenylmonomer, S1–S4 amphiphilic co-polymers have diazine co-monomers such as pyrazine (Pz: S1, S3) and pyrimidine (Pm: S2, S4) for additional hydrogen bonding interactions with plant hormone analytes, such as SA (Fig. 1b). Due to the hydrophobic co-polymer backbone capable of strong π-π interactions with SWNT, highly stable SWNT suspensions (concentrations of 50–75 mg/L) have been generated (Supplementary Fig. 1b).

### Selectivity screening with plant analytes
A CoPhMoRe screen[40] was conducted to investigate the selectivity of the 4 cationic S1–S4 polymer-wrapped SWNTs in detecting 12 key plant hormones and signaling molecules. Out of the 4 polymer-wrapped SWNTs, we found that S3 gave a selective 35% quenching response upon binding of 100 μM SA (Fig. 1c). The list of analytes tested in the selectivity screen includes SA; jasmonic acid (JA), methyl jasmonate (MeJA), gibberellic acid (GA), abscisic acid (ABA), cytokinins: 6-(4-hydroxy-3-methylbut-2-enylamino)purine (zeatin), thidiazuron (TDZ), and 6-benzylaminopurine (BAP), and auxins: 3-indole acetic acid (IAA), 1-naphthaleneacetic acid (NAA) and 2,4-D. $H_2O_2$, the primary ROS signaling molecule, is also added into the list of analytes screened. Their chemical structures are shown in Fig. 1d. The SWNT intensities were measured using a photoluminescence excitation (PLE) spectrometer before and after addition of 100 μM plant hormone analytes, and dimethyl sulfoxide (DMSO) was added as a blank solvent control. Comparatively, S1 and S2 are relatively inert to all the plant hormones (Supplementary Fig. 1c-d) with fluorescence changes of less than ±10% for most plant hormones, except S1 showing moderate quenching response to 2 cytokinins: zeatin (14%) and BAP (11%). S3 upon addition of SA gave high

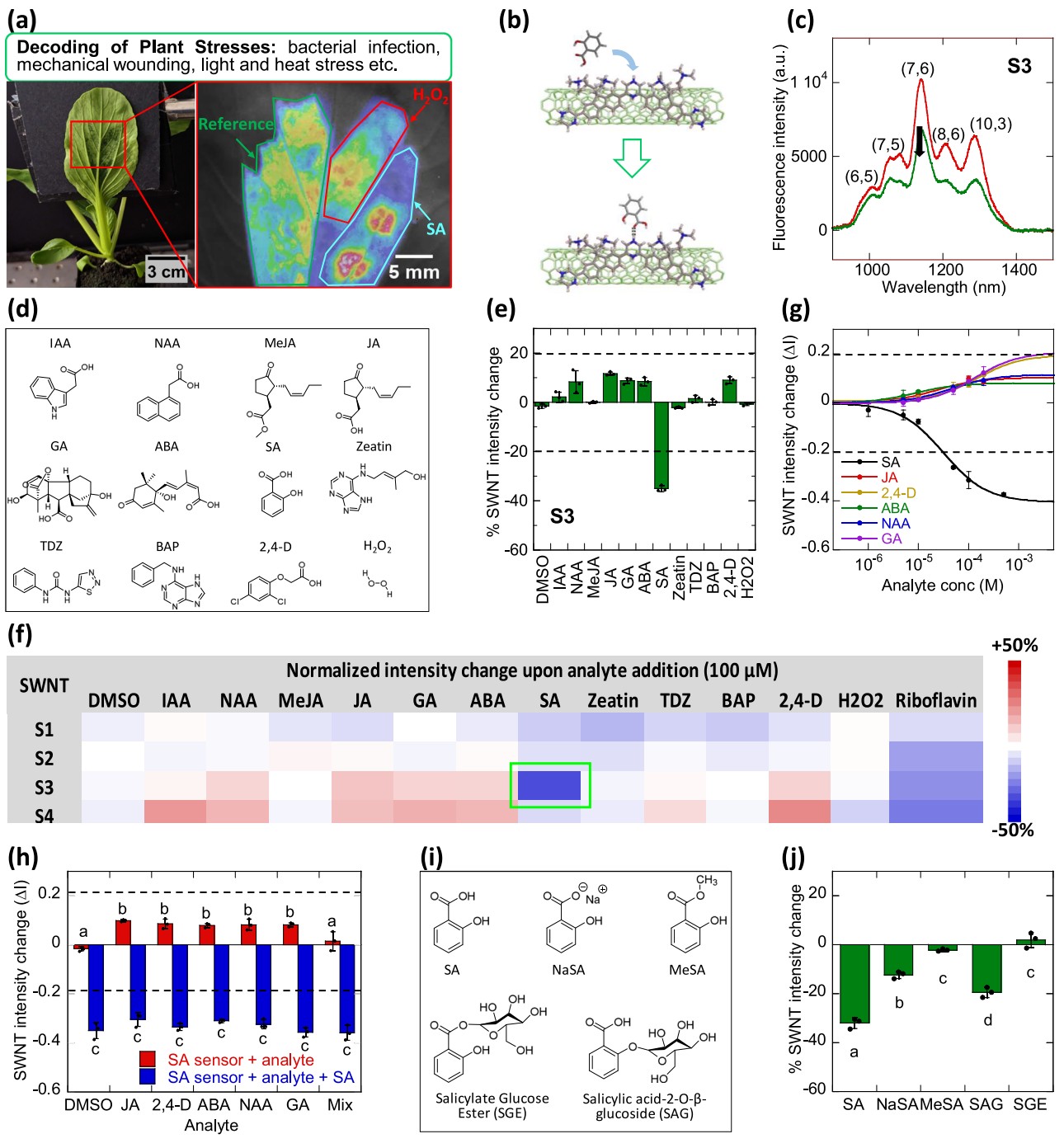

quenching response of 35% and displayed mild turn-on responses of 8-12% for plant hormones JA, ABA, and GA, as well as synthetic auxins (NAA and 2,4-D) (Fig. 1e). S4 on the other hand was lacking in selectivity as it gave non-specific turn-on responses to a number of plant hormones (Supplementary Fig. 1e), including IAA (21%), NAA (15%), 2,4-D (23%), JA (13%), GA (16%), and ABA (14%). The CoPhMoRe screening results are summarized in the heat map shown in Fig. 1f. Besides the 4 polymers, selectivity screening was also done for a fifth CoPhMoRe sensor candidate (S5) using a previously reported aptamer that binds to SA as SWNT wrapping (Supplementary Fig. 2a)[55]. Although a stable SWNT suspension was obtained (Supplementary Fig. 2b), it was found to be non-ideal as a plant nanobionic sensor for SA despite the binding affinity of the aptamer to SA. A detailed description of the selectivity screening results using the aptamer-wrapped SWNT are shown in Supplementary Information and

Supplementary Fig. 2c, d. To summarize, of all the polymer-wrapped SWNTs evaluated, S3 exhibited the highest binding affinity towards SA.

**Characterization of S3 binding to SA**

To study the binding strength of S3 to SA, S3 was titrated with different concentrations of SA ranging from 1 μM to 500 μM. By plotting the nIR fluorescence intensity change against the SA concentrations, a calibration curve can be obtained using the Langmuir adsorption model (Fig. 1g). From the calibration curve, the $K_D$ is derived to be 32 μM. The limit of detection (LOD) of S3 for SA is estimated from the SA concentration needed for a signal-to-background ratio ≥3 and found to be 4.4 μM. S3 reversible binding to SA is further confirmed with a dialysis experiment (Supplementary Fig. 3c). After mixing of S3 with 100 μM SA, the mixture was transferred to a dialysis tube with molecular

**Fig. 1 | Design, screening and characterization of SA sensor. a** Schematic demonstrating the multiplexed sensor platform for the early decoding of different plant stresses using an in planta SA nanosensor (blue) paired with $H_2O_2$ nanosensor (red) and a reference sensor (green); **b** Schematic that demonstrates the binding event of a SA molecule with the cationic polymer-wrapped SWNT that triggers a nIR fluorescence quenching response; **c** S3 SWNT fluorescence spectrum before (red) and after (green) addition of 100 μM SA, giving a 35% quenching response. The magnitude of sensor quenching is obtained by integration of the total sensor intensities from 900 to 1400 nm. Fluorescence peaks of different SWNT chiralities are indicated in parenthesis as (n,m); **d** Chemical structures and abbreviations of plant hormones screened; **e** Sensor fluorescence response to 100 μM plant hormone analytes for S3. Fluorescence quenching or turn-on responses of ±20% (dotted lines) are considered sufficiently large responses due to analyte binding for sensor development. Bar graph shows the mean values with error bars representing standard deviations from independent experiments ($n = 3$). Dots represent each data point. DMSO is used as negative control; **f** heat map summary of the S1–S4 sensor fluorescence response to all the plant hormone analytes with selected SA sensor (S3) highlighted in green. Excitation wavelength: 785 nm; **g** Calibration curve of S3 response (ΔI) against the SA concentration (black) compared against other plant hormone analytes (colored). The SA calibration curve has the $K_D$ of 32 μM ($R = 0.995$) based on the Langmuir adsorption model and a sensor detection limit ~4.4 μM for S/N ratio ≥ 3. Dotted lines at fluorescence responses of ±20% highlight the specificity of S3 binding to SA. Error bars represent the fluorescence quenching responses of $n = 3$ independent replicates for each SA concentration; **h** Fluorescence response of S3 in response to 100 μM plant hormone analytes before (red) and after (blue) addition of 100 μM SA. S3 exhibits preferential binding affinity to SA, showing a strong and consistent quenching response of >30%, even in the presence of other hormones. Bar graph shows the mean values with error bars representing standard deviations from independent experiments ($n = 3$). Dots represent each data point. DMSO is used as negative control. Mix refers to an equimolar mixture of JA + ABA + GA hormone analytes that adds up to concentration of 100 μM; **i** Chemical structures and abbreviations of SA derivatives screened; **j** Fluorescence response of S3 to 100 μM SA and SA derivatives. Fluorescence quenching or turn-on responses of ±20% (dotted lines) are considered sufficiently significant responses due to analyte binding. Bar graphs show the mean values with error bars representing standard deviations from independent experiments ($n = 3$). Dots represent each data point. Different alphabet letters show significant differences using one-way ANOVA with a Tukey's HSD test at $p < 0.05$.

weight cutoff (MWCO) of 3–5 kDa. Within 7 h, S3 fluorescence is found to gradually revert to initial levels as SA desorbs from the S3 into the external dialysate.

As S3 demonstrated mild turn on response to hormones JA, GA, ABA, and synthetic auxins NAA and 2,4D, calibration curves of S3 with different concentrations of these hormones were generated and compared to SA calibration curve. It is clear from these calibration curves that the LOD of the S3 to SA analyte is 3- to 10-fold higher, compared to the other plant hormones (Fig. 1g). Further, we assessed preferential binding of S3 to SA, in the presence of these plant hormones (Fig. 1h). It was observed that even in the presence of interferent plant hormones that gave a mild turn-on response, subsequent addition of SA resulted in a strong and consistent quenching response of >30% similar to the response observed when SA is present alone. Response of S3 to SA in a mixture of plant hormones (JA, ABA, and GA) was also evaluated to mimic in planta conditions. Interestingly, the addition of the hormone mixture mutes the mild turn-on response, and the subsequent addition of SA shows a consistent >30% quenching response. Taken together, these results indicate that S3 shows high preferential binding affinity to SA and will hence be a functional in vivo SA sensor.

To elucidate the mechanism behind the selective binding of S3 to SA, we added riboflavin as a molecular probe that allows us to estimate the SWNT surface coverage of the S1–S4. Out of the 4 SWNTs, S3 had the largest magnitude of nIR fluorescence quenching upon addition of the riboflavin molecular probe. This is indicative of a larger surface area on the SWNT surface available for analyte adsorption[56]. A detailed discussion of the surface coverage results of the polymer-wrapped SWNTs is shown in Supplementary Information and Supplementary Fig. 4.

## SA sensor selectivity against SA derivatives
Like all other biologically active plant hormones, SA is often glycosylated in plants to yield biologically inactive storage forms that are sequestered in the vacuoles. These storage forms are mainly salicylic acid glucoside (SAG) and salicylic acid glucose ester (SGE)[57]. Other common SA derivatives include MeSA, the volatile form of SA that is critical as a mobile SAR signal[18]. Salicylate sodium salt (NaSA), is used mainly for exogenous application to plants due to its high water solubility. These SA derivatives (Fig. 1i) have also been mixed with S3 to test for possible sensor signal interference with SA. It was found that S3 was inert to MeSA and SGE but has moderate quenching responses to NaSA (12%) and SAG (19%) (Fig. 1j). It is hence apparent that the carboxylate group of SA is crucial for electrostatic binding to S3, given that SGE and MeSA with the carboxylate group being glucosidated and

methylated respectively, has relatively muted fluorescence responses. While fluorescence quenching of S3 in response to SA remains highest amongst SA derivatives at 35%, there exists a possibility that the SA sensor could pick up endogenous SAG in plant samples, in addition to free SA. However, the likelihood remains low as our subsequent localization studies have shown that the sensor does not enter vacuoles where SAG is sequestered.

## Subcellular localization of SA sensor in plant cells
The SWNTs are nano in size and functionalized with cationic polymers with high zeta potentials, allowing them to penetrate the cell wall and localize within plant organelles based on the lipid exchange envelope penetration (LEEP) model[58,59]. The SA sensor has a positive zeta potential of +54.1 mV and LEEP model predicts its internalization into cells and cell organelles. Due to conjugated polymer SWNT wrapping, the SA sensor has visible fluorescence attributed to the co-polymer's π–π* band. Hence, its subcellular distribution after infiltration into plants can be monitored using confocal microscopy. SA sensor solutions (1.25 mg/L) were infiltrated into tobacco leaves and the leaves were imaged using confocal microscopy after 1 h. SA sensor fluorescence was observed in the cell periphery of epidermal cells indicating cytoplasmic localization (Fig. 2a–d). To test whether SA sensor localizes to apoplast, plasmolysis was performed. SA sensor fluorescence was seen in the apoplastic space formed by the shrinking protoplast in epidermal cells (Fig. 2e–i). Based on SA fluorescence overlap with the red chlorophyll autofluorescence in mesophyll cells, the sensor also localizes within the chloroplasts (Fig. 2j–m). As SA biosynthesis is mainly localized to chloroplast and cytoplasm, the SA sensor is suitably localized in cells to detect the overall SA production post stress.

## Validation of SA nanosensor in transgenic A. thaliana plants
In plants, SA is produced by two independent pathways, namely the isochorismate (ICS) and the phenylalanine ammonia-lyase (PAL) pathway. The ICS pathway is localized to the plastids and the PAL pathway to the cytoplasm and the contribution of these pathways towards SA production varies in different plants[60–62]. In A. thaliana, the ICS pathway is the dominant pathway, and the first step involves the conversion of chorismic acid to isochorismic acid by isochorismate synthase[63–65]. The SA produced in the chloroplast is subsequently transported to cytoplasm. To determine the in vivo sensitivity range of the SA sensor, an estrogen based chemical inducible XVE system was used to generate transgenic A. thaliana plants accumulating varying levels of SA. The XVE system is a well-established inducible system in plants to control the expression of transgenes[66].

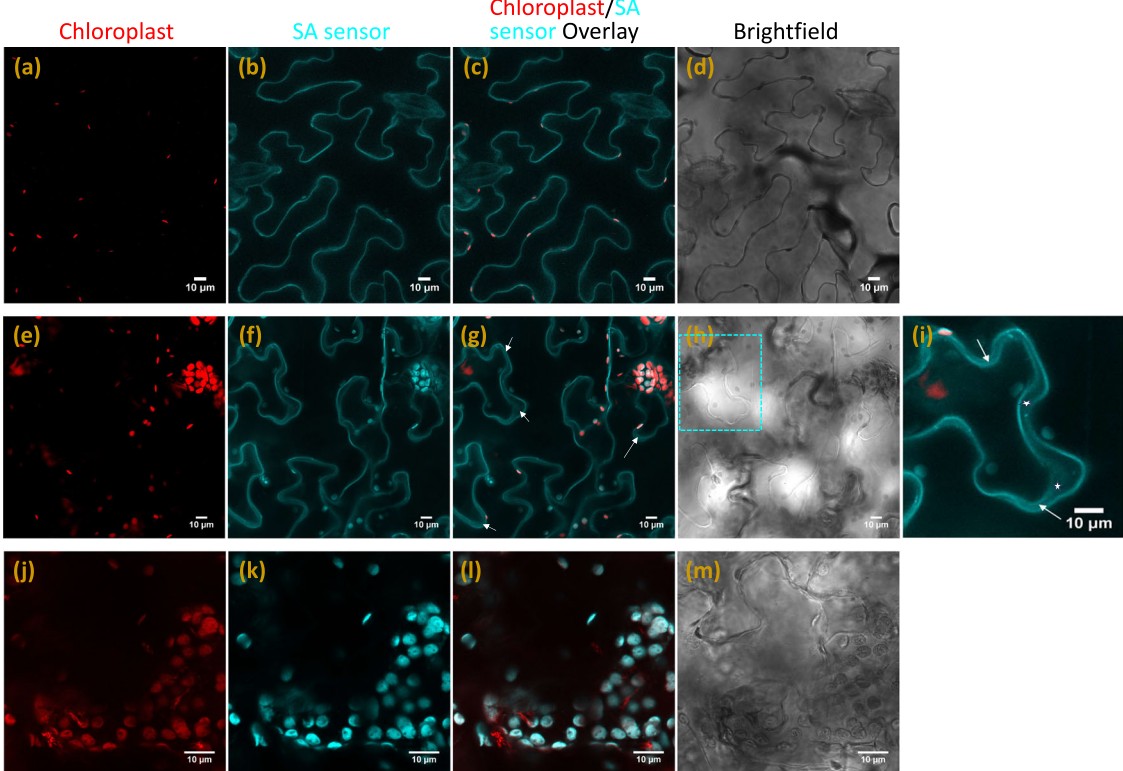

**Fig. 2 | Sub-cellular localization of SA nanosensor in living plants.** Confocal images of tobacco leaf infiltrated with SA sensor to visualize the subcellular localization of SA sensor: **a, e, j** Chlorophyll autofluorescence (red), **b, f, k** SA sensor Cyan florescence, **c, g, l** overlay and (**d, h, m**) brightfield. Row 1–SA fluorescence was observed in epidermal cell periphery indicating cytoplasmic localization. Row 2–plasmolyzed epidermal cells (with 0.8 M Mannitol), showing SA sensor fluorescence in the apoplastic space formed by the shrinking protoplast as indicated by the arrows. **i** Zoom-in overlay image of a cell from Fig. 2g with corresponding brightfield image area marked with dotted cyan box in Fig. 2h, asterisk indicate the apoplastic space. Row 3–overlap of red chlorophyll autofluorescence with cyan SA fluorescence in mesophyll cells indicating chloroplast localization of SA sensor. Row 1, Row 2, and Row 3 Images are representative of at least three independent experiments.

The XVE is activated by the application of inducer estradiol (E2). The *isochorismate synthase 1* (*ICS1*) mutant of *A. thaliana* was complemented with *ICS1* cDNA under the expression cassette of an XVE system vector (pER8-*ICS1*). To investigate the induction rate of *ICS1* transcript, transgenic seedlings were treated with 100 μM of E2 and *ICS1* gene expression was measured at different time intervals ranging from 30 min to 48 h (Fig. 3a). The *ICS1* transcript was detectable at 30 minutes post E2 treatment, increased to highest level at 16 h and then gradually declined in the transgenic plant. Hence, we decided to perform the sensor studies at 16 h time point. Control Wild type and *ICS1* mutant plant*s* did not show increased transcript levels upon induction by E2. Further E2-dose dependent increase in *ICS1* transcript was also evaluated and detected by qPCR at 16 h post treatment with different concentrations of E2 which ranged from 10 μM to 100 μM (Fig. 3b).

To validate the SA sensor in vivo, WT and transgenic plants were treated with different concentrations of E2 at 10 μM, 50 μM, and 100 μM respectively (Fig. 3c). At *t* = 16 h post E2 treatment, SA sensor is infiltrated into the E2 treated leaves on the left of the leaf mid-vein while reference sensor (AT)$_{15}$-SWNT, known to be relatively inert to most plant hormones including SA[38], is infiltrated on the right of the leaf mid-vein (Fig. 3d). From the false-color images obtained at the 16 h time-point, the SA sensor nIR fluorescence appears progressively dimmer as E2 concentrations increase, while the reference sensor fluorescence remains relatively unchanged (Fig. 3e). By taking a ratio of the fluorescence between the SA and reference sensors and normalizing against wild-type *A. thaliana*, we obtain the change in SA concentrations in the inducible SA mutants compared to wild-type, increasing from 11 to 31 and 57 μM for E2 treatment concentrations of 10 μM, 50 μM, and 100 μM respectively (Fig. 3f). This increasing trend is corroborated by LC-MS analysis of leaves at 16 h post treatment that show SA levels increasing from 4.9, to 13.9, and 18.7 μg/g FW for 10 μM, 50 μM, and 100 μM E2 treatment respectively (Fig. 3g, Supplementary Fig. 5). We also investigated the biocompatibility of the SA sensor in *A. thaliana*. As shown in Supplementary Fig. 6, when compared to control water infiltrated plants, the SA sensor-infiltrated plants showed no changes in the chlorophyll content of the leaves, no signs of premature senescence and no difference in the overall growth of the plant.

## Real-time detection of SA production upon bacterial infection in pak choi

SA is the key hormone which mediates plant responses upon pathogen infection[14]. To demonstrate the species independent feature of the sensor, we performed infection studies in a non-model plant, pak choi, which is an important Brassica vegetable crop. Black rot disease, caused by the gram-negative bacterium *Xanthomonas campestris pv. campestris* (*Xcc*) is a serious problem in all Brassica vegetables[67,68], resulting in enormous crop yield losses. Similar to *A. thaliana*, we examined the biocompatibility of the SA sensor in pak choi plants and found no adverse effects on leaf life span, chlorophyll content and overall plant growth (Supplementary Fig. 6).

To probe SA's role in *Xcc* infection in pak choi plants, SA and reference sensors were first infiltrated to the left and right of the pak choi leaf mid-vein respectively approximately in the middle of the leaf (Fig. 4a), followed by *Xcc* infection at *t* = 10 min on the top half of the leaf above the sensor spot. The sensor intensities obtained over time are normalized by the initial intensities prior to *Xcc* infection. The SA sensor fluorescence begins to quench ~1 h post infection, indicative of increasing SA accumulation whereas the reference sensor fluorescence

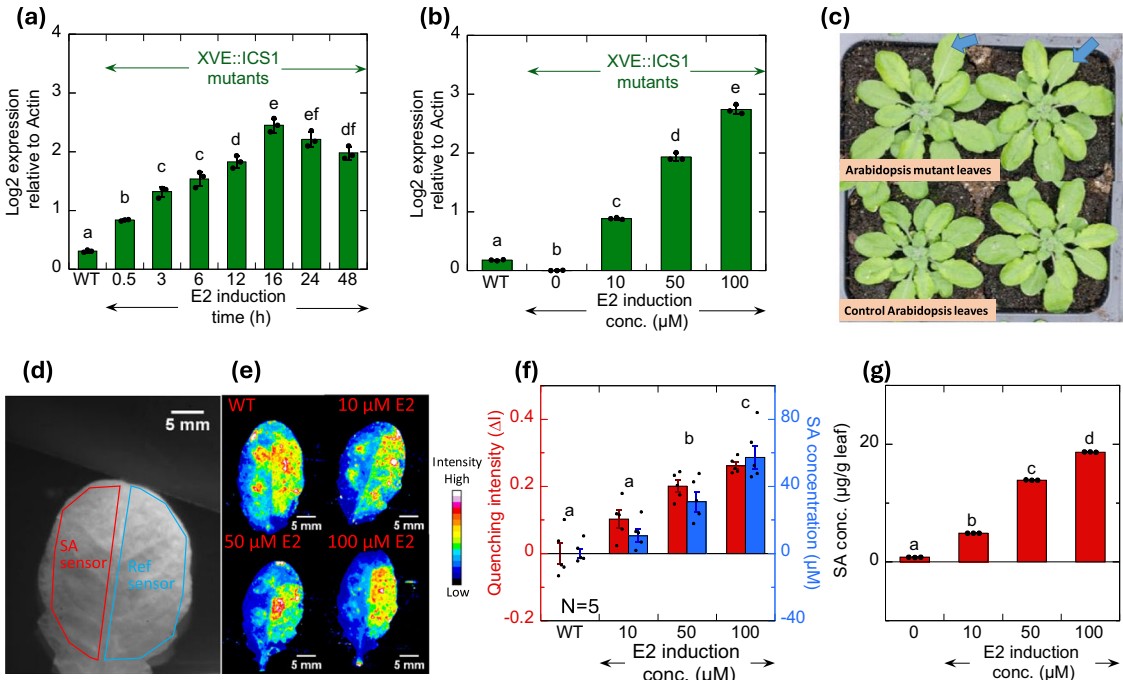

**Fig. 3 | Complementation of *A. thaliana ICS1* mutant plants with *ICS1* gene driven by an inducible XVE expression system to generate transgenic plants with varying amount of SA hormones and in planta validation of SA nanosensor using transgenic *A. thaliana* mutant line (*XVE::ICS1*), where SA levels can be activated and controlled by the amount of estradiol (E2) inducer treatment.** **a** Time dependent expression of *ICS1* gene upon E2 induction, highest gene expression was detected at 16 h post induction. Bar graph show the mean values with error bars representing standard deviations from biologically independent experiments ($n = 3$). Dots represent each data point; **b** increased gene induction with increasing amount of E2 at 16hrs post induction. Bar graph show the mean values with error bars representing standard deviations from biologically independent experiments ($n = 3$). Dots represent each data point; **c** *A. thaliana* mutants (top) and wild-type *A. thaliana* (bottom) leaves subjected to E2 treatment (blue arrows) of different concentrations of 10 μM, 50 μM, and 100 μM; **d** Brightfield image of *A. thaliana* leaf infiltrated with SA sensor and reference sensor on the left and right side of the midvein respectively; **e** False-color images showing nIR fluorescence of sensor-infiltrated wild-type *A. thaliana* and *A. thaliana* mutant leaves, 16 h after E2 treatments of 10 μM, 50 μM, and 100 μM, compared against wild-type *A. thaliana*; **f** nIR fluorescence quenching intensity ratio (red) of SA and reference sensors, and the corresponding local concentration of SA (blue) detected 16 h after 10 μM, 50 μM, and 100 μM E2 treatment compared against wild-type *A. thaliana*. Bar graphs show the mean values with error bars representing standard error from independent experiments ($n = 5$). Dots represent each data point.; **g** Average concentrations of SA in *A. thaliana* determined by LCMS, 16 h after 10 μM, 50 μM, and 100 μM E2 treatment, compared to wild-type. Bar graph show the mean values with error bars representing standard deviations from biologically independent experiments ($n = 3$). Dots represent each data point. Different alphabet letters show significant differences using one-way ANOVA with a Tukey's HSD test at $p < 0.05$.

remains relatively invariant. The magnitude of fluorescence quenching could be converted to local SA concentrations changes using the SA sensor calibration curve. In doing so, concentration maps showing SA spatial distribution in the infected pak choi leaves from 1 – 6 h post infection are obtained, where brighter blue regions correspond to higher SA concentrations (Fig. 4b). The SA concentration map illustrates that certain regions of the sensor spot area has accumulated higher levels of SA while other regions have negligible SA. At $t = 6$ h, the local SA concentration averaged across the entire sensor spot for 3 independent plant replicates reaches 6.6 μM (Fig. 4c). Correspondingly, buffer infiltration did not induce any significant fluorescence changes for both SA and reference sensors within the same measurement period. These results are corroborated by LC-MS analysis of pak choi leaves 6 h post infection with *Xcc*. Buffer-infiltrated leaves show an SA level of 0.58 μg/g FW at 6 h post infiltration while *Xcc* infected leaves exhibit higher SA levels at 0.95 μg/g FW (Fig. 4d, Supplementary Fig. 7). The derivative of SA concentration-time curve further provides real-time information on the rate of SA production in infected leaves, showing an increasing rate of SA production peaking at approximately 2–3 h post *Xcc* infection, following which the rate slows down and stabilizes by 5 h post *Xcc* infection (Fig. 4e). We show, for the first time, real-time spatiotemporal monitoring of SA accumulation in *Xcc* infected pak choi plants, as part of an immune response to the infection. The method used to calculate in planta SA concentrations, and

conversion of SA sensor intensity maps to SA concentration maps are explained in detail in Supplementary Information and Supplementary Fig. 8.

### Sensor-based screening of chemical elicitors of SAR

Exogenous application of SA and its synthetic analogs are known to activate SAR and induce defense priming in plants[69]. However, SA undergoes rapid glycosylation and shows phytotoxicity, which prevents its use as an effective plant protective agent. Efforts are being made to identify eco-friendly and stable chemicals that can activate SA signaling and initiate SAR in crops[70]. The SA sensor developed can be used to rapidly screen the dosage and efficacy of such chemicals and their molecular mechanisms. Pipecolic acid (Pip) is a known activator of plant defense priming and exogenous application of Pip leads to early SA accumulation in tobacco plants[71,72]. We tested whether exogenous application of Pip could increase the levels of SA in pak choi plants. Prior to this, SA sensor was first shown to be inert to addition of Pip (Supplementary Fig. 9a). SA and reference sensor were then infiltrated into pak choi leaves and the same leaf was treated with 1 mM Pip 15 mins later (Fig. 4f). After 1.5 h of Pip treatment, a gradual fluorescence quenching is observed in the SA sensor spot while the reference sensor spot remains unchanged. This fluorescence quenching intensity was converted to the local SA concentration maps shown in Fig. 4g. At 6 h, the local SA concentration, detected by averaging across the entire

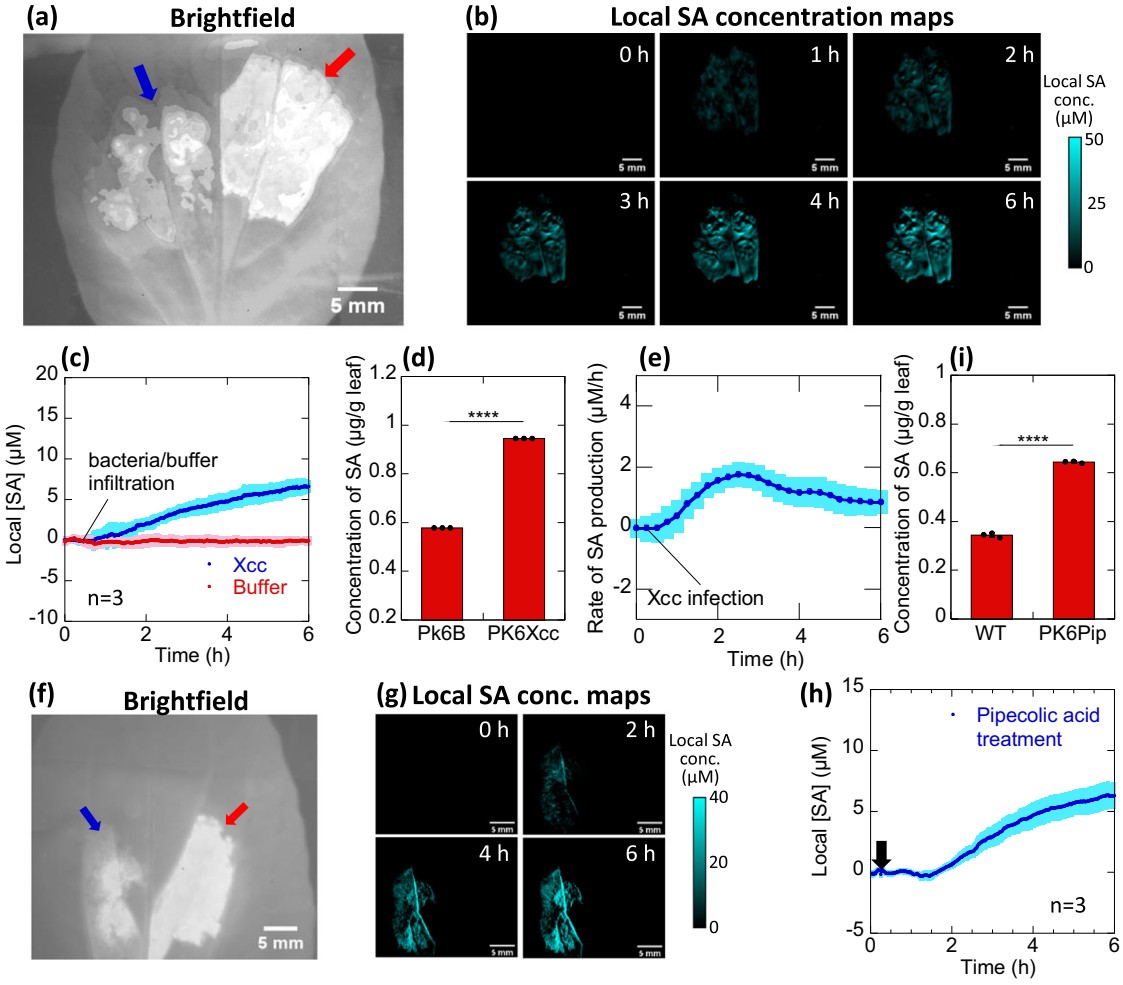

**Fig. 4 | Real-time sensing of a *Xcc* infection event in planta in pak choi.**
**a** Brightfield image of pak choi leaf infiltrated with SA sensor (blue arrow) and reference sensor (red arrow) on the left and right side of the midvein respectively with sensor areas represented as overlay; **b** Derived SA concentration maps after *Xcc* infection at $t = 15$ min on the top half of the leaf above the sensor spot, showing the spatial distribution of SA accumulated within the sensor spot over 6 h; **c** Change in local SA concentrations for *Xcc*-infected (blue) and buffer-infiltrated (red) pak choi plants over 6 h. *Xcc* infection or buffer infiltration of the pak choi leaves occurs at $t = 15$ min. Shaded regions represent standard error across three biologically independent replicates; **d** Average concentrations of SA in pak choi determined by LCMS, 6 h after *Xcc* infection compared to buffer infiltrated pak choi. Bar graph shows the mean values with error bars representing standard deviations from biologically independent experiments ($n = 3$). Dots represent each data point; **e** Rate of SA production derived from the concentration-time curve by taking the gradient of the tangent at every 15-min time-point post *Xcc* infection. Shaded regions represent standard error across three independent replicates; **f** Bright-field images of pak choi infiltrated with the reference sensor and SA sensor to the right and left of leaf midvein respectively with sensor areas represented as overlay; **g** Change in local SA concentrations in pak choi plants before pip treatment (0 h), and after pip treatment (2, 4, and 6 h); **h** Change in the local SA concentration measured upon 1 mM pip treatment at $t = 15$ min (black arrow). Shaded regions represent standard error across three independent replicates; **i** Average concentrations of SA in pak choi determined by LCMS, 6 h after pip treatment compared to WT. Bar graphs show the mean values with error bars representing standard deviations from biologically independent experiments ($n = 3$). Dots represent each data point. Two-tailed unpaired *t*-test, ****$P < 0.0001$.

SA sensor spot, reaches 5.0 μM (Fig. 4h). These results are also corroborated by LC-MS analysis of pak choi leaves 6 h post Pip spray. Control wild-type leaves sprayed with water show an SA level of 0.35 μg/g FW while Pip treated leaves after 6 h post Pip treatment exhibit a higher SA level of 0.65 μg/g FW (Fig. 4i and Supplementary Fig. 9b).

## Multiplexed ROS and SA detection upon biotic and abiotic stresses of pak choi

To demonstrate the ability of multiplexed nanosensors in plant stress elucidation, we paired the SA sensor with a previously published $(GT)_{15}$-SWNT-based nanosensor that selectively detects the key ROS molecule, $H_2O_2$[38]. To investigate if during multiplexing, the sensors can move or stay restricted to their respective infiltrated locations in the leaf, confocal imaging of the SA and Cy3-tagged $(GT)_{15}$-SWNT ROS sensor was performed post infiltration in pak choi leaves. As shown in Supplementary Fig. 10, the fluorescence from SA and ROS sensor was found to remain restricted to their respective infiltrated regions and no mixing was observed. A detailed discussion of the confocal imaging results is provided in Supplementary Information.

For simultaneous and real-time monitoring of SA and ROS production in pak choi leaves post varied stress, both $H_2O_2$ and SA sensors were infiltrated in the leaf together with reference sensor. Derived $H_2O_2$ and SA concentration maps obtained for the multiplexed nanosensors, as well as their respective brightfield images, are shown after the pak choi leaves were subjected to mechanical wounding (Fig. 5a and Supplementary Movie 1), *Xcc* infection (Fig. 5b and Supplementary Movie 2), high light (Fig. 5c and Supplementary Movie 3) and high heat (Fig. 5d and Supplementary Movie 4) stresses. A distinct pattern of ROS and SA generation was observed for each type of stress. To highlight the distinct primary ROS waves induced by each stress, the $H_2O_2$ concentration maps are shown at 5-min time-points for the first 45 min post stress. Subsequently, $H_2O_2$ concentration maps are shown at 1 h,

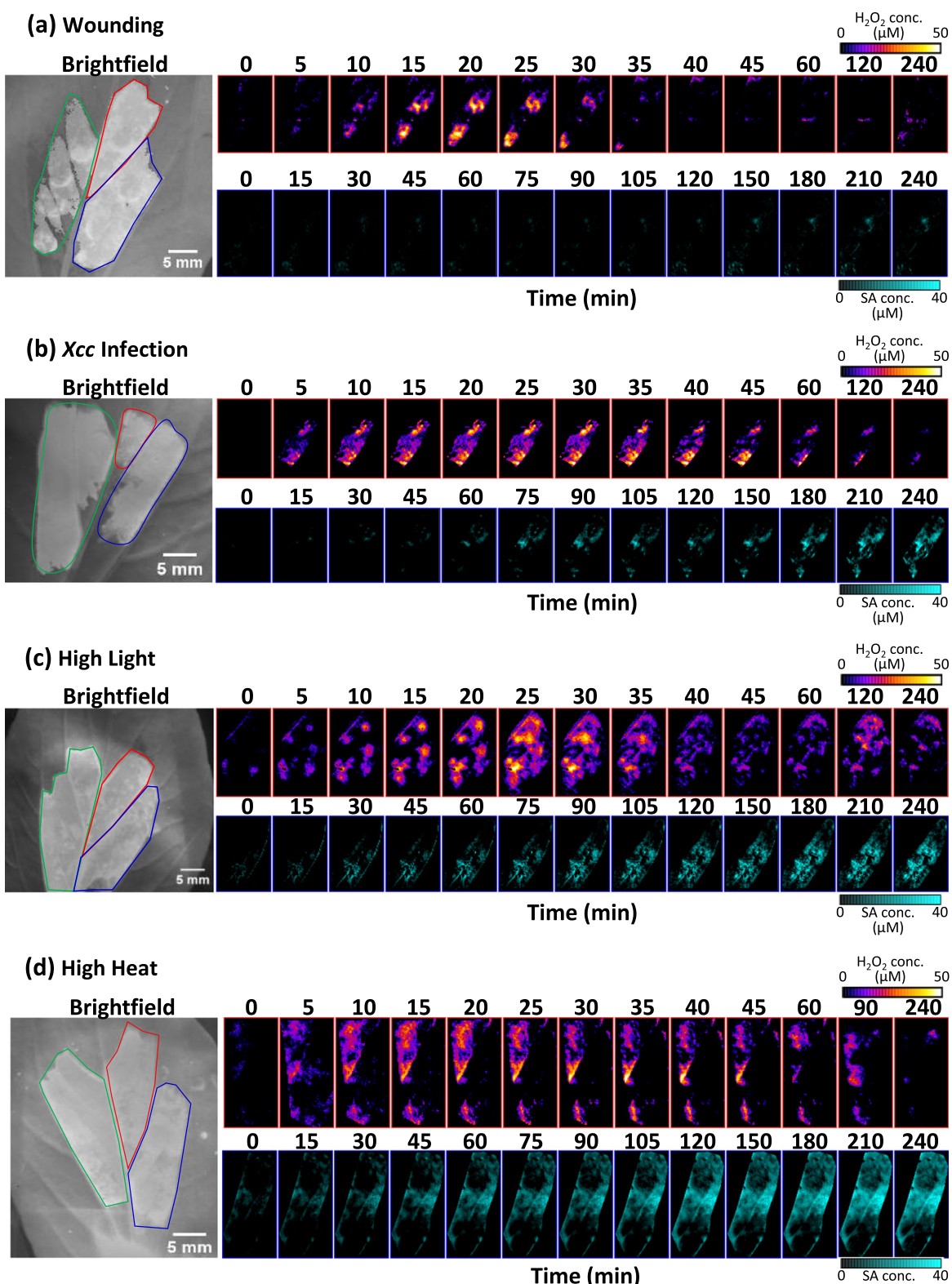

**Fig. 5 | Bright-field and corresponding H₂O₂ (top row) and SA (bottom row) concentration maps of pak choi infiltrated with reference sensor (green), SA sensor (blue), and H₂O₂ sensor (red) under 785 nm laser excitation.** Pak choi plants were subjected to (**a**) mechanical wounding, (**b**) *Xcc* infection, (**c**) high light, and (**d**) high heat treatment respectively. Snapshots of H₂O₂ concentration maps are shown at 5-min intervals for the first 45 min post-stress capturing the first ROS wave, followed by 1 h, 1.5, or 2 h, and 4 h time-points capturing the secondary ROS wave. Snapshots of SA concentration maps are shown at 15-min intervals for the first 2 h post-stress, followed by 30-min intervals for the next 2 h.

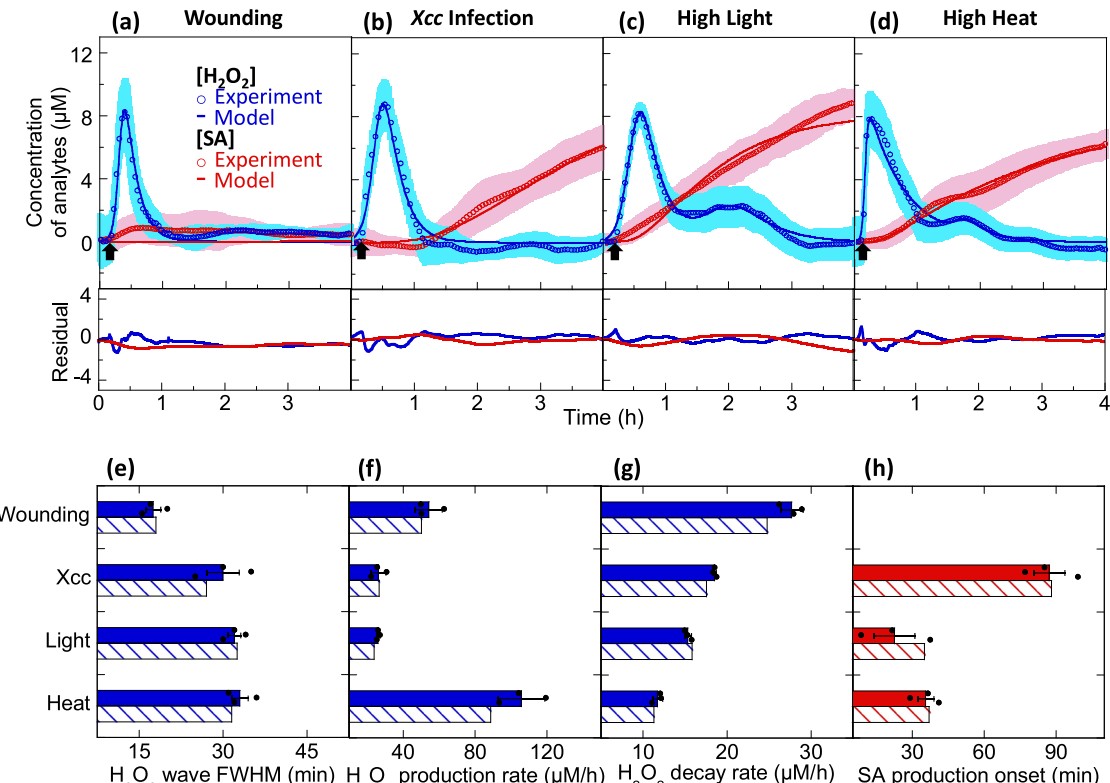

**Fig. 6 | Distinct stress-induced ROS and SA waves in living plants.** Time-plots of $H_2O_2$ concentration derived from nanosensor experiments (blue circles) and mathematical modeling (blue lines), as well as SA concentration derived from nanosensor experiments (red circles) and mathematical modeling (red lines), for pak choi plants that are (**a**) mechanically-wounded, **b** *Xcc* infected, **c** high light treated, and (**d**) high heat treated. Each type of stress treatment of the pak choi leaves occurs at approximately $t = 10$ min (black arrow). Shaded regions represent standard error across 3 independent replicates. Residual plots of $[H_2O_2]_{model}-[H_2O_2]_{experimental}$ (blue) and $[SA]_{model}-[SA]_{experimental}$ (red) are shown underneath each fit for the different plant stresses; Temporal characterization of experimental $H_2O_2$ wave (blue block) and modeled $H_2O_2$ wave (blue diagonal stripes) in terms of (**e**) FWHM, **f** $H_2O_2$ production rate, and (**g**) $H_2O_2$ decay rate for each individual type of plant stress; **h** Onset time of SA production from experimental data (red block) and mathematical modeling (red diagonal stripes) for each individual type of plant stress. Bar graphs of experimental data show the mean values with error bars representing standard error from independent experiments ($n = 3$). Dots represent each data points.

1.5 h/2 h and 4 h time-points to highlight the dynamics of secondary ROS waves induced by heat and light stress. To highlight the distinct onset of SA production induced by each stress, the SA concentration maps are shown at 15-min time intervals for the first 2 h post stress where onset of SA production occurs except for mechanical wounding, followed by 30-min time intervals for the next 2 h. These concentration maps reveal the coordination between these two pathways during the respective stress responses.

By integrating these $H_2O_2$ and SA concentrations across the sensor spot, we obtain time plots showing that for all types of stress, distinct ROS and SA waves are observed. In general, for all stresses applied, appearance of the $H_2O_2$ wave happens within minutes and reaches its maximum followed by recovery within the first hour. SA on the other hand has a longer lag time from the point of stress, followed by gradual accumulation over the 4 h time period (Fig. 6a–d). As a control, we show that neither $H_2O_2$ nor SA are detected by the sensors upon buffer infiltration into pak choi leaves (Supplementary Fig. 11). ROS wave velocities were calculated using the same approach as reported in previous work[38] and found to be dependent on type of stress applied (Supplementary Fig. 12a). ROS wave velocity was highest for high heat treatment at 1.20 cm/min, while similar ROS wave velocities of 0.19 and 0.20 cm/min were obtained for wounding and *Xcc* infection respectively. High light treatment has the lowest ROS wave velocity at 0.08 cm/min.

Our findings from multiplexed sensor reveal the variations in the onset of local SA production relative to the immediate $H_2O_2$ stress signal, indicating that the plant initiates diverse stress responses prior to the decay of the $H_2O_2$ waveform. From the experimental $H_2O_2$

nanosensor time plots, we extracted key characteristics of the ROS wave, including the FWHM defined as the time between the two amplitude midpoints (Fig. 6e), the initial rate of ROS accumulation (Fig. 6f), and rate of ROS decay (Fig. 6g). *Xcc* infection, high light, and high heat treatment produce similar ROS waves with FWHM of 30-33 min. On the other hand, wounding triggers a much faster ROS wave with a FWHM of 17 min, approximately half that of the other stress-induced ROS waves. The wounding-induced ROS wave also follows the typical asymmetric wave with relatively high $H_2O_2$ production and decay rates of 54 µM/h and 28 µM/h respectively. High heat treatment induces ROS waves with a much higher $H_2O_2$ production rate of 106 µM/h coupled with a slow $H_2O_2$ decay rate of 12 µM/h, resulting in a highly asymmetric ROS wave. Out of the 4 stress treatments, *Xcc* infection and high light treatment are difficult to distinguish with the $H_2O_2$ wave alone, as they have similar $H_2O_2$ production and decay rates of 26 µM/h and 15-18 µM/h respectively.

Highlighting the importance of sensor multiplexing in stress elucidation, *Xcc* infection and high light stress are discernible by their subsequent onset of SA production, defined as the time-point at which fluorescence intensity deviates from the standard deviation (Fig. 6h). *Xcc* infection triggered a much later onset of SA production at 87.0 min compared to high light (22.5 min) and high heat (35.5 min). The onset of SA production upon *Xcc* infection coincides with the recovery of the infection-induced ROS wave, indicating that the ROS wave probably precedes SA formation during biotic stress. This is distinct from abiotic stresses represented by high heat and high light treatments that induced SA production earlier, while the ROS waves are still ongoing.

Interestingly, mechanical wounding triggered negligible SA production by 4 h. The dynamics of the SA wave differ from those of the ROS wave because SA does not follow an autocatalytic production mechanism as the ROS wave does. Instead, generation of SA appears to be prompted by ROS. While the generation or first appearance of SA travels as a wave, the subsequent SA concentration profile evolves via a diffusive mechanism. We can hence track the SA source and estimate the SA wave velocity by taking a ratio of the stress application site distance to the sensor spot, and the mean SA onset time (Fig. 6h). Using this approach, the SA wave velocity is calculated to be highest for high light treatment at 0.044 cm/min, followed by high heat treatment at 0.028 cm/min and lowest for Xcc infection at 0.011 cm/min (Supplementary Fig. 12b). With insights gained from the experimental analysis of distinct stress-induced ROS waves, we have formulated a simple mathematical model to elucidate the observed dynamics of SA (see below section).

To summarize, we demonstrate that even though both ROS and SA are generated by different stresses, the ROS and SA waves for each individual stress are different, possibly contributing towards establishing specificity during stress signaling. The fact that we can clearly observe separate timing for SA generation for each type of stress further confirms that the multiplexed nanosensors do not diffuse. In particular, for *Xcc* infection, no fluorescence changes were observed in the SA nanosensor area while the ROS wave is on-going.

### Formulation of model for stress-generated ROS and SA waves

Notably, $H_2O_2$ is one of the earliest stress signals in plants, and concentration-time plots from our experiments (see above "Results" section) suggest that the varying temporal characteristics of the $H_2O_2$ waveform observed within ~1 h may contain sufficient information for the plant to mount an appropriate stress-dependent response. However, the indiscriminate nature of $H_2O_2$ as a reactant leaves the underlying mechanisms of the differential signaling responses and $H_2O_2$ waveforms unresolved. It is suggested that stress-generated ROS can affect the redox state of mutiple proteins, like enzymes, receptors, and transcription factors which can alter, activate and integrate different stress-responsive pathways[8]. Further, ROS is regulated by a network of enzymatic and non-enzymatic scavengers that influences ROS accumulation[73-75]. On the other hand, both the ICS and PAL pathways for SA biosynthesis in plants are regulated by multiple transcription factors in a spatial and temporal manner[76-78]. Hence, in our proposed chemical mechanism, multiple regulatory pathways for $H_2O_2$ and SA exist to impart flexibility in $H_2O_2$ waveforms and SA onset. The integration of different stress signals and their transduction mechanisms through the plant remain an important question in the field. To probe at these open questions, we propose a general chemical mechanism to describe the stress-dependent $H_2O_2$ and SA signatures observed.

$$A + P \xrightarrow{k} A + A, \tag{1}$$

$$A \xrightarrow{k_d} B, \tag{2}$$

$$A + pF \xrightarrow{k_{F_1}} F_1, \tag{3}$$

$$F_1 \xrightarrow{k_{F_2}} F_2 \xrightarrow{k_{F_3}} \dots \xrightarrow{k_{F_n}} F_n \xrightarrow{k_s} S. \tag{4}$$

$$A + pI \xrightarrow{k_I} I, \tag{5}$$

$$I + pF \xrightarrow{k_r} xF_1. \tag{6}$$

Here, $A$ is $H_2O_2$, $P$ is a precursor for $H_2O_2$ such as respiratory burst oxidase homologs (RBOHs), $B$ is a generic degradation product of $H_2O_2$, $I$ is a generic inhibitor of SA, and the $F$ species represent biosynthetic precursors to SA. Species starting with $p$ represent direct precursors, and $xF_1$ represents deactivated $F_1$ that will not proceed in the SA biosynthesis pathway. The $k$ terms are corresponding reaction rates. Equations 1 and 2 comprise the reaction network we previously used to describe the $H_2O_2$ waveform induced upon mechanical wounding[38,79,80], whereby the dominant mechanism of $H_2O_2$ production is via RBOHs. SA production is described in Eqs. 3–4, and Eqs. 5–6 describe a regulation mechanism. In Eqs. 3 and 5, $A$ is involved in a step in the biosynthetic pathway of $S$ and $I$. The production of $I$ inhibits an earlier step in the biosynthesis of $S$ (i.e., $pF$), thus making the relative production rates of $I$ and $pF$ crucial in determining the onset of $S$ production relative to $A$. This model effectively captures the key temporal features of the experimental $H_2O_2$ and SA waves (see Supplementary Information for fitting details). The chemical species are left general, as they may represent different molecules depending on the stress type.

Indeed, recent literature has highlighted the plasticity of systemic ROS signaling in plants[10]. For example, Fichman et al. observed tissue specificity in ROS wave propagation for biotic stress, light stress, and wounding[81]. Beltrán et al. proposed the existence of "sensory" plastids, raising the idea that the subcellular site of stress perception could uniquely affect signal initiation and propagation[82]. If $H_2O_2$ stress signaling or perception can occur in physiologically distinct regions of the plant cell and/or in distinct cell types, then the relative rates of the model reactions could conceivably be tuned by different initial concentrations of $P$, $pF$, and $pI$. ROS production in different subcellular compartments during stress is proposed to contribute towards the specificity of oxidative signaling. The information encoded by ROS signatures is detected by specific ROS sensors to activate stimulus specific responses[83]. Recently, the first cell surface receptor of $H_2O_2$, *hydrogen-peroxide-induced $Ca^{2+}$ increases* (HPCA1) was identified which plays an important role in sensing $H_2O_2$ generated in the apoplast[84]. Other organellar-specific ROS sensors or tissue-specific ROS sensors can also exist. Further, $H_2O_2$ perception can also occur through redox modifications of proteins. The reactivity of cysteine within the ROS sensitive proteins is known to be affected by pH. Cysteine oxidation occurs at pH levels higher than their pKa. Biotic and abiotic stresses are known to bring transient changes to apoplast pH which are different in magnitude and duration, contributing towards specificity in ROS signaling[83]. The molecular mechanisms by which changes in cellular ROS production function to control hormone production need more elucidation. Overall, the model provides a potential unifying mechanism by which stress specificity can be captured by the distinct temporal features of the local $H_2O_2$ waveform, including FWHM, rate of ROS accumulation and ROS decay, shortly after stress perception.

Besides the primary ROS burst produced early within minutes upon stress perception, later ROS bursts are also reported due to activation of early defense signaling networks[8]. These secondary/tertiary ROS bursts are known to happen at time points ranging from hours to days post stress[85]. In this study, mild secondary ROS bursts were observed at 2 and 1.5 h post high light and high heat stresses respectively, while no secondary ROS bursts were observed post biotic stress within the 4 h time period. The majority of biphasic ROS bursts reported and characterized are in response to biotic stress, and are not well described in response to abiotic stresses[83]. Upon biotic stress, the initial ROS burst is reported to happen rapidly within a few minutes, while the second massive burst occurs hours later and plays an important role in cell death[83,86,87]. The secondary ROS burst upon biotic stress in our study probably occurs after the 4 h time period. The molecular mechanisms involved in the generation of the secondary ROS are not well elucidated. A few studies have shown the involvement of RBOH in biphasic ROS production during biotic stress[88,89]. A recent

study also suggests the involvement of other factors contributing to secondary ROS production[90]. As an initial hypothesis, we have modeled the secondary ROS bursts using the same RBOH-dependent mechanism as the initial burst. The modeled data fits well with the observed experimental concentration-time curves for high light and high heat treatments, suggesting the feasibility of this mechanism. However, the precise molecular mechanisms and timings of secondary ROS bursts for different stresses will need further biological investigation to improve the secondary ROS burst modeling process in future studies.

Studies in plant immune response have also shown the existence of antagonism betweeen SA and JA pathways which is important to mount pathogen-appropriate defense responses. SA pathway is induced against hemi-biotrophic/biotrophic pathogens, such as *Xcc*, while JA is produced in response to necrotrophic pathogens and wounding. JA signaling molecules can inhibit SA accumulation and vice versa[91–93]. Kinetics of SA-JA signaling has been shown to be important for the outcome of JA-SA interactions in *A. thaliana* defense response[94]. In wounded leaves of *A. thaliana*, the level of JA increases within minutes[95], and a recent work shows that the rapid accumulation of JA is independent of transcription and translation and may depend on the activity and stability of a key pathway protein[96]. Activation of early JA signaling upon wounding can act as an inhibitor for SA production as observed in our study. There is some evidence that indicates that both SA and JA biosynthesis pathways can be regulated by glutathione (GSH), which is a major antioxidant present in plant cells, but the exact molecular mechanisms are unknown. GSH plays a vital role in mediating stress responses by determining the redox state of the cell and can link ROS to SA and JA production[80,97,98]. The well-established SA-JA antagonism is, however, stress specific, as under high light conditions their interaction appears to be synergistic[99]. There might be other hormones or metabolites that can act as inhibitors of SA synthesis under defined stress conditions coordinated by ROS. Hence, in our model, we have left *I* as a generic inhibitor of SA synthesis in plants.

The multiple intermediates involved in producing $S$ ($pF$, $F_1$, and $F_2$ and $F_n$) are essential for describing the experimental data of the *Xcc* infection response, where the onset of $S$ occurs after the decay of $A$. This delay in the onset of $S$ relative to $A$ implies the existence of intermediate chemical species prior to the production of $S$. We have arbitrarily chosen a small number of intermediates for the model reaction network. Besides transcriptional activation of SA biosynthesis by the ICS or PAL pathways, SA can also be rapidly re-converted from its conjugated storage forms, such as SAG and SGE depending on plant needs[100]. Hence, the intermediate steps involved in generating SA can vary for different stresses. Relatively earlier onset of SA accumulation after high light and high heat stress is indicative of release of SA from conjugated forms instead of transcriptional activation of the biosynthesis pathway from the beginning. Overall, despite the simplicity of our proposed model, the results can provide a reasonably accurate description of the stress-specific onset of SA production observed experimentally.

## Discussion

We developed a CoPhMoRe-based nanosensor for real-time and non-destructive detection of an important plant hormone, SA, and demonstrated its use in both model (*A. thaliana*) and non-model (pak choi) plants. The nanosensor shows excellent selectivity to the SA hormone and is biocompatible for plant studies. The sensor was validated using well-characterized *A. thaliana* mutant plants defective in SA synthesis and transgenic *A. thaliana* plants accumulating varying levels of SA. Further studies were performed in the non-model plant pak choi, an important vegetable crop, to highlight the species-independent feature of the CoPhMoRe nanosensors[5]. The SA sensor enabled monitoring of real-time changes in SA production both

spatially and temporally in pak choi during plant pathogen interaction. Temporal analysis revealed the non-uniform rate of SA production after infection with rapid SA accumulation within 2–3 h post *Xcc* infection, which later declines and stabilizes by 5 h. The sensor allowed us to perform quantitative kinetic analysis of SA induction post pathogen infection that will aid in further understanding of SA regulation and its physiology. Compounds known as plant activators are being developed that can mimic SA to confer disease resistance in plants. The sensor can also potentially facilitate rapid screening of such chemicals, such as Pip, that primes plant through the SA pathway, as well as determine the effective concentrations required.

Crosstalk between plant hormones and their signaling pathways forms the core of plant growth and stress response. A rapid increase in ROS production known as 'the oxidative burst' is a common plant response to biotic and abiotic stress conditions[10]. It has been proposed that the specificity of ROS signaling leading to an appropriate stress response depends on multiple factors such as sub-cellular production site, signal intensity, and interactions with other signaling molecules[85,101]. Through multiplexing of $H_2O_2$ and SA sensors, the interplay between these molecules in response to different stress conditions could be detected in real-time in pak choi plants. Our study revealed that each stress stimulus generated distinct temporal patterns of ROS and SA production. Heat stress is a major abiotic stress affecting crop productivity. Heat stress promotes the accumulation of SA in plants leading to basal thermotolerance[102,103]. Increased SA levels under high temperature is proposed to regulate the antioxidant defense systems and improve the photosynthetic efficiency leading to thermotolerance[104]. Exogenous application of SA provides thermotolerance to plants by lowering the ROS induced oxidative stress[102,105,106]. SA is also reported to play a role in plant acclimation to high light[107]. Similar to heat stress, SA alleviates high light induced damage in plants by activating the antioxidant system and protecting the photosynthetic machinery[108]. On the other hand, it has been reported that during defense responses against pathogens, increase in SA levels are preceded by apoplastic $H_2O_2$ bursts[12]. Recent studies suggest that during plant response to biotic stress, the position where SA interacts with ROS signaling and ROS with SA signaling depends on the pathosystem and the origin of ROS[109]. In our study using pak choi plants, the multiplexed sensors reveal that both high heat and high light stimuli resulted in production of SA but at an earlier time point than the infection-triggered SA generation in pak choi. Under heat and light stresses, SA production occurs while the ROS burst is still ongoing. Under infection stress, SA production was observed after ROS wave recovers. Hence the SA wave dynamics induced by abiotic and biotic stress are different. The ROS waves of heat and light stress are also distinct, with high heat treatment producing a highly asymmetric ROS wave with the fastest initial $H_2O_2$ accumulation out of the 4 stresses, followed by a gradual decay. They also produce secondary ROS waves at 2 h and 1.5 h respectively post-stress. Wounding in plants can be due to both biotic and abiotic causes. SA generation in response to wounding appear to vary in plants. In rice and peas, an early decrease in endogenous SA content was observed after wounding[110,111]. In *A. thaliana*, an increase in SA was observed after 6 h post mechanical wounding[112]. Similarly, in pak choi, upon mechanical wounding, our sensors revealed the absence of SA involvement in the initial local response for the first 4 h of monitoring. Mechanical wounding also produced ROS waves with the smallest FWHM, directly correlating to the shortest duration of ROS burst.

As such, our study had revealed, for the first time, the interconnection and hierarchy between $H_2O_2$ and SA molecules to different stress stimuli in real time. These experimental insights allowed us to generate a chemical kinetics model that effectively captures the temporal features of each stress-specific wave, forming the basis of early plant stress decoding. The plant stress response involves activation and coordination of multiple signaling pathways. The differences in

timing and levels of these signaling molecules production are proposed to have an impact on the efficacy of the stress response[113–115]. Essentially, encoded within the $H_2O_2$ waveform is the type of stress that the plant had just experienced, triggering distinct downstream stress-specific signaling pathways. This study paves the way for multiplexing of many CoPhMoRe nanosensors for simultaneous detection of various plant analytes following an external stimulus with the use of microneedle arrays embedded with different sensors. As we multiplex more sensors and monitor the plant analytes over longer periods of time in vivo, it will be essential to evaluate the life span, stability, as well as diffusion, of nanosensors within plant cells over days. Confinement of the nanosensors within separate microneedles can help prevent sensor diffusion and interference. Future work with the nanosensors could also include the study of plant hormone production rates within various cell organelles using nIR confocal microscopes, to attain insights into the dynamic changes of various plant hormones within distinct organelles. This would revolutionize the way we examine signaling pathways and gain appropriate time resolution. The ability to unravel the sequential events occurring in cells with spatial-temporal perspective during the time course of the stress response through multiplexing will enhance our knowledge about various signaling network activated in plants especially during conditions of multiple stress combinations. The data obtained can be used to further strengthen our mathematical model to plant responses to elucidate various stresses in crops more accurately. This will help in developing strategies to improve plant stress tolerance, and mitigating crop losses due to environmental stress. Nanobionic plants embedded with the multiplexed sensors can serve as sentinel plants, facilitating early asymptomatic detection of plant stress within hours enabling timely responsive measures to minimize yield loss. So far, studies attempting the simultaneous detection of multiple analytes in plants, by generating transgenic plants expressing various genetically encoded biosensors, have been few and far between[116,117]. This is largely because the generation of transgenic plants is time-consuming and many commercial crops either remain recalcitrant to established transformation protocols or display low transformation efficiency. By contrast, nanosensors can be easily introduced by infiltration into any plant species bypassing the bottleneck posed by transformation.

# Methods

## Materials

All reagents, catalysts, and solvents were purchased from Sigma-Aldrich Ltd and Tokyo Chemical Industry Ltd, unless otherwise stated. Plant hormone analytes used for CoPhMoRe library screening were purchased from Sigma-Aldrich unless otherwise stated and used as received. Storage forms of SA (SAG and SGE) were purchased from Toronto Research Canada while zeatin was purchased from Gold Biotechnology, Inc.

## Synthesis of fluorene-diazine co-polymers

For synthesis of PFPz co-polymers, S1 (para linkages) and S3(meta linkages), equimolar quantities of F-diEs and Pz-diBr were added to a 20-mL microwave-safe vial, together with Pd(dppf)Cl$_2$ catalyst (3 mol %). The vial was crimp sealed, pumped down to vacuum and backfilled with Ar thrice. Degassed anhydrous THF:DMF (2:1) was added to the vial to dissolve the monomers. Na$_2$CO$_3$ solution (5 eq) in degassed deionized water was then added into the vial. The resultant reaction mixture was heated in a microwave reaction chamber at 130 °C for 15 min. Upon cooling to room temperature, the mixture was precipitated in water and obtained by filtering through a 0.45 μm nylon filter. The PFPz co-polymer was then washed with water until the washing was neutral (pH = 7) to remove all Na$_2$CO$_3$. The polymer solids were redissolved in THF and filtered again through a 0.45 μm nylon syringe filter to remove other undissolvable impurities. To the filtered polymer solution, sodium diethyldithiocarbamate solution in methanol (100 mg/mL) was added (100 eq) and stirred for 3 h to remove Pd catalyst residues. The purified PFPm co-polymer was retrieved again by re-precipitation and repeated washing in water to remove excess sodium diethyldithiocarbamate and the associated Pd complexes. The same polymerization and purification procedures apply to synthesis of the PFPm co-polymers, S2 (para linkages) and S4 (meta linkages) with Pm-diBr monomer in place of Pz-diBr.

## Cationic functionalization of fluorene-diazine co-polymers

The neutral PFPz and PFPm co-polymers (100 mg) are dissolved in THF (4 mL) in a 20-mL microwave-safe vial, followed by addition of 1 M HCl (4 mL). Upon addition of HCl, the polymer solution turns cloudy and partially redissolves when deionized water is added (12 mL). The solution is crimp-sealed and heated in a microwave at 120 °C for 5 min. Upon cooling to RT, the dissolved cationic polymer solution is filtered through a 0.45 μM nylon syringe filter before transferring to a cellulose ester dialysis tube (8-10 kDa MWCO). The polymer solution is dialyzed with deionized water for 2 days with the external dialysate refreshed every 8 h, to remove all HCl, THF, and oligomers. Finally, the cationic polymer solids of S1–S4 were retrieved via freeze-drying.

## Suspension of SWNT using cationic co-polymers

For preparation of S1–S4 wrapped SWNT, 2.5 mg of cationic polymer was dissolved in 1 mL deionized water, before addition of 1 mg of HiPCO SWNT (Nanointegris). The mixtures were sonicated with a 3-mm probe tip (Qsonica Q500) for 30-min at 20% amplitude in an ice bath. The resultant S1–S4 wrapped SWNT suspensions were subjected to ultra-centrifugation at 153,145 g for 2 h to remove SWNT aggregates. For preparation of SA aptamer S5 wrapped SWNT, 100 μL of SA aptamer solution (100 μM in 0.1 M NaCl) was diluted in 775 μL of MES buffer, 10 mM (pH 5.5). Separately, 6,5-enriched Comocat SWNT (Signis® SG65i) was mixed in deionized water at 2 mg/mL and 125 μL of SWNT was added to the diluted SA aptamer solution. The mixture was subjected to tip sonication for 15-min at 20% amplitude in an ice bath. The resultant S5-wrapped SWNT suspension was subjected to 2 rounds of 90 min centrifugation at 16,000 g to remove SWNT aggregates. The concentrations of S1-S5 SWNT suspensions were determined by UV-Vis spectrophotometry (Agilent Cary 5000) using SWNT absorbance at 632 nm with an extinction coefficient of 0.036 (mg/L)$^{-1}$cm$^{-140}$.

## Suspension of SWNT using single-stranded DNA oligomers

DNA oligomers were purchased from Integrated DNA Technologies. HiPCO SWNTs were purchased from Nanointegris while (6,5)-enriched ComoCAT SWNTs were purchased from Sigma-Aldrich. To generate DNA-wrapped SWNTs, 1 mg of single-stranded DNA and 0.25 mg of SWNT were mixed in 1 ml of 50 mM NaCl. (GT)$_{15}$ was used in suspension of HiPCO SWNTs for formation of $H_2O_2$ nanosensor while (AT)$_{15}$ was used in suspension of (6,5)-enriched ComoCAT SWNTs for formation of reference sensor. The mixture was sonicated with 3 mm probe tip for 15 min at 22% amplitude in an ice bath. The sample was then centrifuged twice at 16,000 g for 90 min each to remove unsuspended SWNT bundles.

## In vitro screening of plant hormone analytes

Photoluminescence excitation (PLE) measurements, performed on a Jobin-Yvon Nanolog-3 spectrofluorometer coupled with an InGaAs detector, were used for in vitro screening of polymer-wrapped SWNTs against the plant hormone analyte library. 2 mg/L stock suspensions of polymer-wrapped SWNTs were prepared in MES buffer (10 mM MES, 10 mM MgCl$_2$, pH 5.5) and added to a four-sided quartz cuvette. All analytes except $H_2O_2$ were dissolved in DMSO at 50 mM while 3% hydrogen peroxide was diluted to 50 mM in deionized water. 2 μL of each analyte was added to 998 μL of SWNT suspension, to obtain final analyte concentration of 100 μM. Samples were excited at wavelength 785-nm, with band-width of ±25 nm while nIR emission was collected in

the wavelength range of 900-1600 nm. Integration time of each emission scan was 30 s. The total SWNT fluorescence counts obtained from the spectra were integrated across 900-1400 nm to account for all major SWNT chiralities. Total SWNT fluorescence was measured before and after addition of plant hormone analytes and fluorescence intensity change is calculated by taking a ratio of the two. For 2D excitation-emission map, the excitation wavelength is tuned at 5 nm steps from 500 to 800 nm, while SWNT fluorescence is generated across 900-1300 nm at each step. Integration time of each scan was 30 s, and an entire map was generated within 30 min.

## Plant materials and growth conditions

*A. thaliana* WT and mutant plants were grown for five to six weeks at 22 °C with 60% relative humidity in long-day conditions (16 h light/8 h dark) under white light at 100 μmol m$^{-2}$ s$^{-1}$ in a growth chamber. Pak choi plants were grown for two weeks in a controlled environment chamber under long days (16 h) and 70% humidity. *A. thaliana ICS1* mutant (SALK_111380C) was obtained from *A. thaliana* Biological Resource Centre (ABRC).

## Sub-cellular localization of SA sensor using confocal microscopy

After synthesis of SA sensor, S3 free polymer that is unbound to SWNTs were removed by centrifugal filtration using the Amicon® Ultra-4 Centrifugal Filter tubes with 100 kDa MWCO. Complete removal of free polymer is confirmed by the absence of the polymer UV-Vis absorbance band ($\lambda_{max} = 370$ nm) in the filtrate. The SA sensor solution was then diluted to 1.25 mg/L and infiltrated into *N. benthamiana*/pak choi leaves. Infiltrated plants were kept in dark at 28 °C for 1 h. Plasmolysis of leaf was induced by treatment with 0.8 M Mannitol solution (Sigma-Aldrich, Singapore). All leaf samples were viewed with an inverted confocal microscope (SP8, Leica, Germany). For confocal imaging, the excitation laser and detector wavelength ranges used for each fluorophore is as follows, SA sensor: 405 nm (415−440 nm); Cy3-(GT)$_{15}$ SWNT: 552 nm (565−580 nm); Chlorophyll autofluorescence: 594 nm (640−700 nm). Confocal images were analyzed using the ImageJ software.

## Detection of SA in plants using SA nanosensor

A needleless 1-mL syringe was used to infiltrate sensors into the abaxial side of *A. thaliana* and pak choi leaves. The active and reference SA sensors were infiltrated on each side of the leaf midveins. All SWNTs were diluted to 3 mg/L in deionized water and infiltrated with gentle pressure applied on the other side of the leaf. This is to minimize tissue damage due to sensor infiltration. All in vivo measurements were done 0.5 h after infiltration of SWNTs. Intact *A. thaliana* and pak choi plants were placed in front of the near-infrared imaging standoff system which consisted of a liquid-N$_2$-cooled Princeton Instruments NIRvana InGaAs detector coupled to a Nikon AF-S Micro-NIKKOR 105-mm f/2.8D lens. The infiltrated leaf was secured with tape at a distance of about 1 m from the camera lens, with an 830-nm long-pass filter (Semrock Razor Edge LP02-830) placed in front of the camera lens. 785-nm laser with incident power of 32 mW/cm$^2$ was used as excitation source with exposure time of 30 s per frame. The laser excitation spot on the leaf is circular with radius of 12.3 mm, covering both sensor infiltration sites on both sides of the leaf mid-vein. The intensity counts of SWNT fluorescence were integrated over the entire sensor spot. For multiplexed detection of SA and H$_2$O$_2$ in plants using respective nanosensors, the reference sensor (AT)$_{15}$-SWNT that was inert to both SA and H$_2$O$_2$ was infiltrated on the left side of the leaf midvein. On the other hand, H$_2$O$_2$ sensor and SA sensors were both infiltrated on the right side of the leaf midvein, separated by a secondary vein. For ratiometric sensing, the ratio of integrated SWNT fluorescence intensities of active and reference sensors were taken, which reduces errors due to leaf movement or laser amplitude instability over time.

## Generation of XVE-induced A. thaliana transgenic plants

The $_p$ER8 vector was a generous gift from Prof Chua Nam-Hai's lab. It uses a highly inducible XVE system which can be activated by E2. The AtICS1 coding region was PCR-amplified from *A. thaliana* and then inserted it into the expression cassette of an XVE system vector ($_p$ER8-*ICS1*). This vector was introduced into Agrobacterium tumefaciens strain EHA105 and *A. thaliana ICS1* mutant plants were transformed by floral dip transformation protocol. Transformants were selected on Hygromycin plates and T3 generation seeds were used for all induction experiments.

## E2 induction studies

For selection of transgenic plants showing optimal induction and gene expression studies, 2-weeks-old plants were transferred to media containing different concentrations of E2 (10, 50, and 100 μM) and subsequently incubated for 16 h before RNA extraction. For sensor in vivo validation, the plants were grown in soil (4 weeks), then induced by applying E2 at different concentrations (10, 50, and 100 μM) onto the leaves using a paint brush, and incubated for 16 h before sensor infiltration.

## RNA extraction and quantitative RT-PCR

Total RNA was extracted from leaves of mutant plants using RNeasy® Plus Mini kit from Qiagen (Singapore). About 1 μg total RNA was used for first-strand cDNA synthesis. Reverse transcription and quantitative RT-PCR were using iScipt RT Supermix (Bio Rad, Singapore) and KAPA SYBR FAST qPCR Master Mix (2X) (Sigma-Aldrich, Singapore), respectively using primer pair (5′ CGTCGTTCGGTTACAGGTTC 3′ and 5′ AGAAGATCGGGACGACCAAC 3′) for *ICS1*. Data analysis was performed using Relative Quantitation software from ABI using $2^{-\Delta\Delta CT}$ method.

## Xcc infection of pak choi

For growing Xanthomonas, YGC media was prepared by mixing glucose (10 g/L), yeast extract (10 g/L) calcium carbonate (20 g/L), and Agar (10 g/L) in sterile water and autoclaved at 121 °C for 15 minutes. For inoculum preparation, a single colony from master plate was and inoculated in 10 mL of YGC broth and incubated at 28 °C in a shaking incubator (200 rpm) for 24 h. The turbidity of the broth was measured with a spectrophotometer (Amersham Ultrospec 2100 pro) at a wavelength of 600 nm and the suspension was diluted to reach an absorbance of 0.5 OD, which corresponds to a concentration of approximately $5 \times 10^8$ cfu/mL. Pak choi plants at 2-weeks-old stage were first infiltrated with SA and reference sensors 30 min prior to imaging with the nIR camera. At $t = 10$ min of measurement, the plants were inoculated with *Xcc* on the top half abaxial side of the fourth leaf using needleless syringe above the sensor spots. Control plants were inoculated with YGC broth in the same location on the leaf to mimic the effect of mechanical damage done to the leaf during inoculation.

## Mechanical wounding, light stress, and heat stress treatments

The different stress treatments were performed as follows. Mechanical wounding was inflicted ~1 cm above the sensor spot using a sharp toothpick across the midrib of the pak choi leaf. High light stress was performed by exposing the tip of the pak choi leaf to high intensity light (3000 μmol m$^{-2}$ s$^{-1}$) for 5 min using a cold fiber optic light source (Zeiss CL 1500 ECO). A localized heat stress treatment was conducted by placing the end of a stainless-steel metal rod that is pre-heated in a water bath to 50 °C in contact with the tip of the pak choi leaf for 30 s.

## Biocompatibility study of SA sensor

SA sensor (2 mg/L) was infiltrated into *A. thaliana* and pak choi leaves. As control, water was infiltrated. Both sensor and water-infiltrated control plants were grown for 4 weeks and the leaves/plants were monitored for any growth defect and early senescence. The

chlorophyll content of sensor and control water infiltrated leaves was measured using a leaf Spectrometer (CI-710s Spectra Vue, CID-Bioscience, USA)

## Chemical treatment of pak choi leaves

To detect SA changes upon Pip treatment, pak choi plants at 2-weeks-old stage were first infiltrated with SA and reference sensors 30 min prior to imaging with the nIR camera. At $t = 10$ min after imaging begins, exogenous application of 1 mM Pip, dissolved in water, was carried out using paint brush and applied evenly on the surface of the leaf.

## Image analysis

Image and data analysis were performed using ImageJ and Matlab R2021a. Time plots of the nanosensor responses to various stresses and chemical treatment are normalized to initial values prior to respective treatments at $t = 10$ min. For analysis of $H_2O_2$ sensor, Matlab R2021a was used to isolate the 200 pixels from the sensor ROI with highest fluorescence quenching response. For analysis of reference and SA sensors, ImageJ was used to obtain the average fluorescence quenching across the respective sensor ROIs. The normalized intensities are divided with the reference sensor intensity in order to obtain a ratiometric response that accounts for leaf movement and laser intensity drifts. The ratiometric data was then smoothed with a moving average filter using a window width of 10 time points. Concentration maps could then be derived from the fluorescence intensity changes using the in vitro sensor calibration curve. Detailed methodology of SA concentration calculations is elaborated upon in Supplementary Information.

## LC and sample preparation

For hormone measurement, 110 mg of E2 treated, *Xcc* bacteria infected, pip treated, and control plant leaf material from *A. thaliana* or pak choi was harvested into a 2 ml microfuge tube. 400 µL of 10% methanol containing 1% acetic acid was added to each tube and ground in a bead beater (Qiagen) with 3 mm tungsten beads at 25 Hz/s for 3 min. Samples were placed on ice for 30 min followed by centrifuging at 13,000 × *g* for 10 min at 4 °C. Supernatant was transferred to a new 2 ml microfuge tube. In all, 400 µL of 10% methanol containing 1% acetic acid was added to the pellet and repeated the extraction once more. Supernatant was pooled and filtered using 0.22 µm centrifuge tube filter (Corning, USA) and used for hormone measurements. The extracts were concentrated under the flow of nitrogen and 2 µl of concentrated extracts were injected into Accucore RP-MS C18 column (2.1 × 100 mm, 2.6 µm particle size; Thermo-scientific, Breda, Netherlands) with 0.3 mL/min flow rate for LC-ESI-MS/MS analysis of SA independently. The mobile phase (A) is composed of 0.1% formic acid in water and 100% methanol constituted the mobile phase (B). The elution gradient started with 5% methanol for 1–5 min, then increased linearly to 100% at 10–18 min and then decreased to 5% at 20.0 min. Oven temperature was maintained at 40 °C. Detection was performed in a targeted selected ion monitoring mode on the Thermo Orbitrap Q–Exactive quadrupole mass spectrometer (Thermo-scientific Singapore). The MS parameters included drying gas temperature, 350 °C; gas flow rate, 12 L/min; nebulizer pressure, 35 psi; sheath gas temperature, 400 °C; sheath gas flow rate, 12 L/min; delta electron multiplier volt, 500 V; and capillary voltage, 4000 V in the negative ionization mode. The AGC target was 2E5. The mass resolution was then set at 70k for MS1 and 17.5k MS2 scan. Xcalibur software (version 5.0, Thermo) was used to control the instrument and to acquire and process the MS data. Trace finder and free style software (version 5.0, Thermo) was used for pre-processing the raw data and to determine the analyte peak area response relative to external standard. A calibration curve with a series of analytical standards of SA was used for quantification. Analytes were then quantified based on the standard

curve to determine the concentrations. Three independent biological replicates were analyzed in each treatment and control group for both *A. thaliana* and pak choi.

## Reporting summary

Further information on research design is available in the Nature Portfolio Reporting Summary linked to this article.

## Data availability

The authors declare that all data supporting the findings of this study are available within the paper and any raw data can be obtained from the corresponding author on request. Source data are provided with this paper.

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

## Acknowledgements

This research was supported by the National Research Foundation (NRF), Prime Minister's Office, Singapore, under its Campus for Research Excellence and Technological Enterprise (CREATE) program. The Disruptive & Sustainable Technology for Agricultural Precision (DiSTAP) is an interdisciplinary research group of the Singapore-MIT Alliance for Research and Technology (SMART) Center. TKP is grateful for support from the National Science Foundation Graduate Research Fellowship Program under Grant No. 2141064. The mathematical and analytical work is supported by Nanotechnology for Agricultural and Food Systems (A1511) [grant no. 2021-67021-33999/project accession no. 1025638] from the USDA National Institute of Food and Agriculture.

## Author contributions

M.C.Y.A., J.M.S., S.R., and M.S.S. wrote the manuscripts. T.K.P, J.C., and S.W. edited the manuscripts. M.C.Y.A. designed the sensors. J.M.S. designed, performed, and analyzed all plant-related experiments and confocal microscopy experiments. M.C.Y.A., S.M., and S.I.L. designed, performed, and analyzed all sensor-related in vitro and in planta experiments. T.K.P. designed and formulated all mathematical models for $H_2O_2$ and SA waves. D.T.K. designed and performed SA aptamer-wrapped SWNT experiment. S.S. performed and analyzed all LCMS experiments. M.S.S., S.R., G.P.S., and N.H.C. supervised the project. All authors have revised the manuscript and given their approval of the final version.

## Competing interests

The authors declare no competing interests.
