## [Peer Review File · Nature Communications]

Decoding early stress signaling waves in living plants using nanosensor multiplexingReviewer #1 (Remarks to the Author):

In the current paper, the authors have developed a new salicylic acid (SA) nanosensor consisting of cationic polymers and carbon nanotubes and visualized SA dynamics in Arabidopsis and pak choi plants. In combination with a previously-developed H₂O₂ nanosensor, they have shown SA and H₂O₂ responses in planta and in silico under light, heat, pathogen and wound stress conditions. The experimental designs and aims are novel. However, a major shortcoming of this paper would be that the authors fail to present convincing data that show the sub-cellular localization and ligand-specificity of the SA nanosensor in vivo. For instance, the authors claimed that the SA nanosensors are localized to the cytoplasm and chloroplasts; but I feel that the SA nanosensors should remain in the apoplasts due to the infiltration method used to load the nanosensor into plants. If the SA nanosensors are localized to a variety of sub-cellular compartments such as apoplastic, cytoplasmic and chloroplastic regions, I am not sure which SA concentrations are calculated and analyzed in this paper. Moreover, this SA nanosensor is sensitive not only to SA but also to other plant hormones, such as jasmonic acid (JA), gibberellic acid (GA), abscisic acid (ABA) and a synthetic auxin. SA induces 35% reduction in the fluorescent signal of this sensor, whereas JA, GA, ABA and auxin induce approximately 10% increase in the fluorescent signal. Environmental stresses simultaneously change the levels of various plant hormones in vivo. For instance, when a 5% reduction in the nanosensor signal is observed upon stress in plants (Figure 5), is its reduction only due to an increase in SA level or complicated hormone balances (10% increase in SA minus 5% increase in JA, GA or ABA)? Furthermore, this SA nanosensor is sensitive to SA derivatives, such as salicylate sodium salt (NaSA; 12% reduction in the signal), and SA storage forms, salicylic acid glucoside (SAG; 19% reduction in the signal), which make the data (readout of the sensor) difficult to interpret.

A further significant concern would be that the data (images) are not clear. I cannot distinguish the SA and ROS waves claimed by the authors presumably due to the low sensitivity/dynamic range of this sensor. I feel that this problem should be solved by showing high-resolution images and movies.

To strengthen the conclusions of this paper, the following comments should be addressed.

Major comments:

(1) The authors need to show that endogenous SA, but not other plant hormones/SA derivatives, solely changes the fluorescent signals of the SA sensor, by making calibration curves for other factors or analyzing hormone-deficient mutants, such as *aos* or *opr3* mutants for JA in Arabidopsis with the SA nanosensor. One of the big advantages of this new nanosensor is that the authors can promptly conduct these experiments without making transgenic lines. Otherwise, it is difficult to accept the SA concentrations/dynamics calculated from the fluorescent signal of the SA nanosensor.

(2) Page 11, Line 207-209. The authors claim that "The SA sensor, with a highly positive zeta potential of +54.1 mV, was found to be capable of traversing through the cell membrane into the cytoplasm, and further localizing into the chloroplast (Figure 2c-e)." However, (2-1) I am not sure whether the SA sensor is localized to the apoplast and/or cytoplasmic regions. To confirm it, the authors should conduct plasmolysis assay and show the apoplastic fluorescence signals. (2-2) Figure 2 shows hundreds of chlorophyll red autofluorescence and tens of punctate SA sensor "blue" fluorescence. However, there are only a few overlaps between the red and blue signals. Please provide a rationale to claim that the SA sensor is further localized to the chloroplasts. If this claim is true, is the SA sensor localized to the outer or inner membrane, stroma, thylakoid membrane or lumen? Please show higher resolution images or infiltrate the SA sensor into transgenic Arabidopsis leaves expressing fluorescent marker proteins in chloroplasts as conducted in Figure 3.

(3) In Figure 4, please explain the detailed method to calculate the SA concentration using the calibration curve (intensity fold-changes; %) in Figure 1g. Because 1) the SA sensor is localized to the various sub-cellular compartments in which basal SA concentrations are different and 2) the total amount of the SA sensor infiltrated into the plant tissue is not estimated, I do not think that the authors could precisely calculate the SA concentrations in planta.

(4) In Figure 4c and 4d and Line 263-269, the local SA concentration gradually increased from 0 μ M to 6.6 μ M (0.95 μ g/g FW) at 6 h. I was puzzled by these data. 1) The authors showed that the

minimum detection limit of the SA sensor is 4.4 μM (Line 175-177). 2) Given that 4.9 $\mu\text{g/g}$ FW for 10 μM estradiol (Line 224) corresponds to a 0.1-fold quenching change (estimated by Fig 3e), 0.95 $\mu\text{g/g}$ FW should correspond to a 0.02-fold change at the maximum (6 h). I am not convinced that this SA sensor could detect such small amounts of SA (lower than the detection limit; or equal to the noise levels). Indeed, the fluorescent (false color) images are not distinguishable (Figure 4g).

(5) Line 306 and the title, the authors suggest that "distinct ROS and SA waves are observed." However, I cannot follow where the ROS and SA waves occurred in Figure 5 and S7. Please provide their spatial and temporal information including velocities of the ROS and SA waves. Convincing pictures and movies (1 frame per 30 s) would help understand the authors' claim.

(6) Professor Ron Mittler's team reported RBOH-dependent ROS waves upon a variety of environmental stresses including light stress. To support the authors' model, it would be important to test Arabidopsis mutants such as rboh/f mutants. Once again, one of the big advantages of this new nanosensor is that the authors can promptly conduct the experiments without making transgenic lines.

Minor comments:

(1) Appropriate statistical analyses are needed such as regarding the specificity of the SA sensor (Figure 2b) and dose-dependency (Figure 3) to strengthen the conclusions.

(2) Line 375, the authors conclude that "This model (Figure 5c-f) effectively captures the key temporal features of the H₂O₂ and SA waves". However, the model predicts single-peak, transient H₂O₂ dynamics, which is not consistent with the bimodal (two or more-peaks or oscillating) H₂O₂ dynamics observed in high light and heat stress responses (Figures 5e and 5f). The authors need to comment on this.

(3) The SA and H₂O₂ nanosensors and the reference are infiltrated into the same leaf. I was wondering whether these sensors are diffused/transported throughout the leaf and mixed in planta. If they are mixed, the experimental design is incoherent. Please explain how the authors separate the sensors in a leaf.

Minor edits:

1) Line 260, Figure 1f -> Figure 1g?

Reviewer #2 (Remarks to the Author):

Use of nanosensors to detect stress signals to enable early stress detection in plants is of importance for nano-enabled agriculture. Here, the manuscript entitled "Decoding early stress signaling waves in living plants using nanosensor multiplexing" tried to develop and use multiplexing nanosensors to detect H₂O₂ and SA signals in stressed plants simultaneously. Although authors claimed these are multiplexing nanosensors. However, the model build in this study is not well described. Also, the verification and validation of the hypothesized model need more efforts. I would recommend authors to do a substantial revision on the manuscript. Below are some other comments.

1. Figure 2c, it seems that this is not a good layer of chloroplasts from the leaves. Also, for Figure 2e, it seems that this is a combination confocal image including bright field.
2. The compatibility of the used nanosensors in Arabidopsis is not shown.
3. Different nanosensors are injected into the leaf. Whether the nanosensors can move to other leaf areas which might cause confounding effect?
4. Figure 3d. Do we have the images at different time points?
5. Figure 5. H₂O₂ signals are detected in wounding stress, while no detection of SA was found in wounding stress. I am wondering if authors can do the continuous wounding to see if the SA can be detected by the nanosensor or not.
6. The fitting curve of SA in Figure 5 d and of H₂O₂ in Figure 5e is not good, suggesting that there

is room to optimize the model.

7. Whether the organelle distribution of nanosensors could affect its detection efficiency on H₂O₂ and SA? The production rate of H₂O₂ and SA is varied in different cell compartments.

8. The quality of figures should be improved. The resolution is low.

Reviewer #1 (Remarks to the Author):

In the current paper, the authors have developed a new salicylic acid (SA) nanosensor consisting of cationic polymers and carbon nanotubes and visualized SA dynamics in Arabidopsis and pak choi plants. In combination with a previously-developed H₂O₂ nanosensor, they have shown SA and H₂O₂ responses *in planta* and *in silico* under light, heat, pathogen and wound stress conditions. The experimental designs and aims are novel.

[Author response]:

We thank the reviewer for the positive assessment of our novel experimental designs and aims. We have made significant changes to our revised manuscript, together with additional experimental data, to address the concerns raised.

[Reviewer 1 comment 1] However, a major shortcoming of this paper would be that the authors fail to present convincing data that show the sub-cellular localization and ligand-specificity of the SA nanosensor *in vivo*. For instance, the authors claimed that the SA nanosensors are localized to the cytoplasm and chloroplasts; but I feel that the SA nanosensors should remain in the apoplasts due to the infiltration method used to load the nanosensor into plants. If the SA nanosensors are localized to a variety of sub-cellular compartments such as apoplastic, cytoplasmic and chloroplastic regions, I am not sure which SA concentrations are calculated and analyzed in this paper.

[Author response]:

In response to the reviewer's concern on this point, we have now performed additional confocal microscopy (Figure 2) and plasmolysis to demonstrate that the SA sensor post infiltration localizes to cytoplasm, apoplast and chloroplast post infiltration, where SA biosynthesis is reported to happen. The SA concentrations calculated in this paper reflects the overall total SA levels in cells post stress. The manuscript under is revised accordingly in Line 226 as follows.

Line 226: "SA sensor fluorescence was observed in the cell periphery indicating cytoplasmic localization (Figure 2a-d). To test whether SA sensor localizes to apoplast, plasmolysis was performed. SA sensor fluorescence was seen in the apoplastic space formed by the shrinking protoplast (Figure 2e-i). Based on SA fluorescence overlap with the red chlorophyll autofluorescence, the sensor also localizes within the chloroplasts (Figure 2j-m). As SA biosynthesis is mainly localized to chloroplast and cytoplasm, the SA sensor is suitably localized in cells to detect the overall SA production post stress.

Figure 2: Confocal images of tobacco epidermal cells to visualize the subcellular localization of SA sensor : (a, e, j) Chlorophyll autofluorescence (red), (b, f, k) SA sensor Cyan florescence, (c, g, l) overlay and (d, h, m) brightfield. Row 1 – SA fluorescence was observed in cell periphery indicating cytoplasmic localization. Row 2 – fluorescence of cells plasmolyzed with 0.8 M mannitol, SA sensor fluorescence observed in the apoplastic space formed by the shrinking protoplast as indicated by the arrows. (i) Zoom in image of a cell from panel f, (*) indicate the apoplastic space. Row 3 – Overlap of red chlorophyll autofluorescence with cyan indicating chloroplast localization of sensor.

We have also previously shown that the infiltration method does not restrict the nanosensors to apoplast. As demonstrated in Lew, T.T.S. et al, *Nature Plants* 6 (2020), 404–415, the nanosensors are able to move into the cytoplasm and cell organelles. These sensors are nano in size and functionalized with cationic polymers with high zeta potentials, allowing them to penetrate the cell wall and localize within plant compartments based on the lipid exchange envelope penetration (LEEP) model. The LEEP model described in Lew, T. T. S. et al, *Small* 14 (2018), 1802086, predicts that sensors will be able to cross membranes and localize to chloroplasts depending on their corona surface charge and nanoparticle size. We are also working on an expanded, hyperspectral study of this sensor construct at the subcellular level, comparing the observed localization to theoretical predictions. We have clarified the explanation of LEEP model in the manuscript, Line 218.

Line 218: “The SWNTs are nano in size and functionalized with cationic polymers with high zeta potentials, allowing them to penetrate the cell wall and localize within plant organelles based on the lipid exchange envelope penetration (LEEP) model^{59,60}. The SA sensor has a positive zeta potential of +54.1 mV and LEEP model predicts its internalization into cells and cell organelles.”

[Reviewer 1 comment 2] Moreover, this SA nanosensor is sensitive not only to SA but also to other plant hormones, such as jasmonic acid (JA), gibberellic acid (GA), abscisic acid (ABA) and a synthetic auxin. SA induces 35% reduction in the fluorescent signal of this sensor, whereas JA, GA, ABA and auxin induce approximately 10% increase in the fluorescent signal. Environmental stresses simultaneously change the levels of various plant hormones *in vivo*. For instance, when a 5% reduction in the nanosensor signal is observed upon stress in plants (Figure 5), is its reduction only due to an increase in SA level or complicated hormone balances (10% increase in SA minus 5% increase in JA, GA or ABA)?

[Author response]:

To address this question, we performed additional measurements to confirm the sensor specificity to SA. As shown in Figure 1h, it was observed that even in the presence of interferent plant hormones that gives a mild turn-on response, subsequent addition of SA gives a strong and consistent quenching response of >30% similar to the response when SA is present alone. Statistical analysis (one-way ANOVA with a Tukey's HSD test at $p < 0.05$) showed no significant differences in the magnitude of SA sensor quenching, in the presence or absence of interference plant hormones. Further, we have evaluated the sensor response in the presence of a mixture of interference plant hormones (JA, ABA and GA). Interestingly, addition of the hormone mixture mutes the mild turn-on response, and subsequent addition of SA again shows a consistent >30% quenching response. Hence, this preferential SA binding affinity of the sensor will likely exclude the possibility of other interferent molecules affecting the sensor signal. The manuscript (Line 186) has also been revised to include this additional result.

Line 186: "Further, we assessed S3 preferential binding to SA, in the presence of these plant hormones (Figure 1h). It was observed that even in the presence of interferent plant hormones that gave a mild turn-on response, subsequent addition of SA resulted in a strong and consistent quenching response of >30% similar to the response observed when SA is present alone. S3 response to SA in a mixture of plant hormones (JA, ABA and GA) was also evaluated to mimic *in planta* conditions. Interestingly, addition of the hormone mixture mutes the mild turn-on response, and subsequent addition of SA shows a consistent >30% quenching response. Taken together, these results indicate that S3 shows high preferential binding affinity to SA and will hence be a functional *in vivo* SA sensor."

Figure 1h: Fluorescence response of S3 in response to 100 μM plant hormone analytes before (red) and after (blue) addition of 100 μM SA. S3 exhibits preferential binding affinity to SA, showing a strong and consistent

quenching response of >30%, even in the presence of other hormones. DMSO analyte is used as negative control. Mix refers to an equimolar mixture of JA+ABA+GA hormone analytes that adds up to concentration of 100 μ M. Bar graphs show the mean values with error bars representing standard deviations from independent experiments (n = 3). Dots represent each data point (n = 3). Different alphabet letters show significant differences using one-way ANOVA with a Tukey's HSD test at $p < 0.05$.

[Reviewer 1 comment 3] Furthermore, this SA nanosensor is sensitive to SA derivatives, such as salicylate sodium salt (NaSA; 12% reduction in the signal), and SA storage forms, salicylic acid glucoside (SAG; 19% reduction in the signal), which make the data (readout of the sensor) difficult to interpret.

[Author response]:

With regards to the SA derivatives, salicylic acid glucoside (SAG) is the storage form of SA, which is mainly sequestered in the vacuole. Our localization studies have shown that the sensor is unable to enter the vacuoles, hence SAG should not be detected by the nanosensor within the cell. To our best of knowledge, free SA is not derivatized to salicylate sodium salt (NaSA) in plants, NaSA is a SA analogue, used mainly for exogenous application to plants due to its high water solubility. NaSA was included as a sensor screening target also because we were interested in the potential pharmacological applications, given that NaSA is also a commonly-used anti-inflammatory drug. To summarize, based on the sensor selectivity and sub-cellular localization, we believe that our sensor will largely detect active free SA in plants. The above has been clarified in the main text, Line 204 and Line 212.

Line 204: "Other common SA derivatives include MeSA, the volatile form of SA that is critical as a mobile SAR signal²⁰. Salicylate sodium salt (NaSA), is used mainly for exogenous application to plants due to its high water solubility."

Line 212: "While S3 fluorescence quenching in response to SA remains highest amongst SA derivatives at 35%, there exists a possibility that the SA sensor could pick up endogenous SAG in plant samples, in addition to free SA. However, the likelihood remains low as our subsequent localization studies have shown that the sensor does not enter vacuoles where SAG is sequestered."

[Reviewer 1 comment 4] A further significant concern would be that the data (images) are not clear. I cannot distinguish the SA and ROS waves claimed by the authors presumably due to the low sensitivity/dynamic range of this sensor. I feel that this problem should be solved by showing high-resolution images and movies. To strengthen the conclusions of this paper, the following comments should be addressed.

[Author response]:

In response to the reviewer's concern, we have significantly revised Figure 5 of the manuscript and converted the false-color sensor intensity images to time-dependent H₂O₂ and SA concentration map images using the respective nanosensor calibration curves. The manuscript main text has modified in Line 338. To highlight the distinct primary ROS waves induced by each stress, the H₂O₂ concentration maps are shown at 5-min time-points for the first 45 min post stress. Subsequent, H₂O₂ concentration maps are shown at 1 h, 1.5 h/2 h and 4 h time-points to highlight the dynamics of secondary ROS waves induced by heat and light stress. To highlight the distinct onset of SA production induced by each stress, the SA concentration maps are shown at 15-min time intervals for the first 2 h post stress where onset of SA production occurs with the exception of mechanical wounding, followed by 30-min time intervals for the next 2 h. From these new images, the generation of distinct ROS waves and SA can be clearly observed. Supplementary videos of H₂O₂ and SA concentration maps (Supp. Video 1-4) for each type of plant stress treatment are also added in this revised manuscript submission.

Line 338: "Derived H₂O₂ and SA concentration maps obtained for the multiplexed nanosensors, as well as their respective brightfield images, are shown after the pak choi leaves were subjected to mechanical wounding (Figure 5a, Supp. Video 1), *Xcc* infection (Figure 5b, Supp. Video 2), high light (Figure 5c, Supp. Video 3) and high heat (Figure 5d, Supp. Video 4) stresses. Distinct pattern of ROS and SA generation was observed for each type of stress. To highlight the distinct primary ROS waves induced by each stress, the H₂O₂ concentration maps are shown at 5-min time-points for the first 45 min post stress. Subsequently, H₂O₂ concentration maps are shown at 1 h, 1.5 h/2 h and 4 h time-points to highlight the dynamics of secondary ROS waves induced by heat and light stress. To highlight the distinct onset of SA production induced by each stress, the SA concentration maps are shown at 15-min time intervals for the first 2 h post stress where onset of SA production occurs with the exception of mechanical wounding, followed by 30-min time intervals for the next 2 h. These concentration maps reveal the coordination between these two pathways during the respective stress responses.

Figure 6: Bright-field and corresponding H_2O_2 (top row) and SA (bottom row) concentration maps of pak choi infiltrated with reference sensor (green), SA sensor (blue), and H_2O_2 sensor (red) under 785 nm laser excitation. Pak choi plants were subjected to (a) mechanical wounding, (b) *Xcc* infection, (c) high light and (d) high heat treatment respectively. Snapshots of H_2O_2 concentration maps are shown at 5-min intervals for the first 45 min post-stress capturing the first ROS wave, followed by 1 h, 1.5 or 2 h, and 4 h time-points capturing the secondary

ROS wave. Snapshots of SA concentration maps are shown at 15-min intervals for the first 2 h post-stress, followed by 30-min intervals for the next 2 h.

[Reviewer 1 comment 5] The authors need to show that endogenous SA, but not other plant hormones/SA derivatives, solely changes the fluorescent signals of the SA sensor, by making calibration curves for other factors or analyzing hormone-deficient mutants, such as *aos* or *opr3* mutants for JA in *Arabidopsis* with the SA nanosensor. One of the big advantages of this new nanosensor is that the authors can promptly conduct these experiments without making transgenic lines. Otherwise, it is difficult to accept the SA concentrations/dynamics calculated from the fluorescent signal of the SA nanosensor.

[Author response]:

As suggested by the reviewer, we have generated the calibration curves of the SA nanosensor when other plant hormones such as JA, GA, ABA and synthetic auxins (NAA, 2,4-D) are added, in comparison with SA. As shown in Figure 1g, the sensor response is much higher towards SA compared to the other plant hormones. Generally, the detection limit of a sensor is estimated by having a Signal/Noise (S/N) ratio of ≥ 3 at which the analyte can be reliably detected. At S/N ratio ≥ 3 ($\Delta I \geq \pm 0.05$), the detection limit of the SA sensor is estimated to be 4.4 μM for the SA analyte. Comparatively, the detection limit for other plant hormones are 3-fold to 10-fold lower. This, together with the sensor's preferential SA binding affinity shown in Figure 1h, allows us to be certain that SA alone significantly changes the fluorescence signals of the sensor. The manuscript main text (Line 182) has been adjusted to include these results.

Line 182: "As S3 demonstrated mild turn on response to hormones JA, GA, ABA, and synthetic auxins NAA and 2,4D, calibration curves of S3 with different concentrations of these hormones were generated and compared to SA calibration curve. It is clear from these calibration curves that the LOD of the S3 to SA analyte is 3- to 10-fold higher, compared to the other plant hormones (Figure 1g). Further, we assessed S3 preferential binding to SA, in the presence of these plant hormones (Figure 1h). It was observed that even in the presence of interferent plant hormones that gave a mild turn-on response, subsequent addition of SA resulted in a strong and consistent quenching response of $>30\%$ similar to the response observed when SA is present alone. S3 response to SA in a mixture of plant hormones (JA, ABA and GA) was also evaluated to mimic *in planta* conditions. Interestingly, addition of the hormone mixture mutes the mild turn-on response, and subsequent addition of SA shows a consistent $>30\%$ quenching response. Taken together, these results indicate that S3 shows high preferential binding affinity to SA and will hence be a functional *in vivo* SA sensor.

Figure 1: (g) Calibration curve of S3 response (ΔI) against the SA concentration (black) compared against other plant hormone analytes (colored). The SA calibration curve has the K_D of 32 μM ($R = 0.995$) based on the

Langmuir adsorption model and a sensor detection limit $\sim 4.4 \mu\text{M}$ for S/N ratio ≥ 3 . Dotted lines at fluorescence responses of $\pm 20\%$ highlight the specificity of S3 binding to SA. Error bars represent the fluorescence quenching responses of three replicates for each SA concentration; (h) Fluorescence response of S3 in response to $100 \mu\text{M}$ plant hormone analytes before (red) and after (blue) addition of $100 \mu\text{M}$ SA. S3 exhibits preferential binding affinity to SA, showing a strong and consistent quenching response of $>30\%$, even in the presence of other hormones. DMSO is used as negative control. Mix refers to an equimolar mixture of JA+ABA+GA hormone analytes that adds up to concentration of $100 \mu\text{M}$. Bar graphs show the mean values with error bars representing standard deviations from independent experiments ($n = 3$). Dots represent each data point ($n = 3$). Different alphabet letters show significant differences using one-way ANOVA with a Tukey's HSD test at $p < 0.05$.

[Reviewer 1 comment 6] Page 11, Line 207-209. The authors claim that “The SA sensor, with a highly positive zeta potential of +54.1 mV, was found to be capable of traversing through the cell membrane into the cytoplasm, and further localizing into the chloroplast (Figure 2c-e).” However, (2-1) I am not sure whether the SA sensor is localized to the apoplast and/or cytoplasmic regions. To confirm it, the authors should conduct plasmolysis assay and show the apoplastic fluorescence signals. (2-2) Figure 2 shows hundreds of chlorophyll red autofluorescence and tens of punctate SA sensor “blue” fluorescence. However, there are only a few overlaps between the red and blue signals. Please provide a rationale to claim that the SA sensor is further localized to the chloroplasts. If this claim is true, is the SA sensor localized to the outer or inner membrane, stroma, thylakoid membrane or lumen? Please show higher resolution images or infiltrate the SA sensor into transgenic Arabidopsis leaves expressing fluorescent marker proteins in chloroplasts as conducted in Figure 3.

[Author response]:

In response to the reviewer’s concern on this point, we have now performed additional confocal microscopy (Figure 2) to demonstrate that the SA sensor post infiltration localizes to cytoplasm (Figure 2a-d), apoplast (Figure 2e-i) and chloroplast (Figure 2j-m) post infiltration, where SA biosynthesis is reported to happen. As suggested by the reviewer, plasmolysis assay was performed. The cells were treated with mannitol to induce plasmolysis. As shown in Figure 2e-i, the SA sensor fluorescence can be observed in the apoplastic space formed by the shrinking protoplast as indicated by the arrows. To confirm chloroplast localization, higher resolution images chloroplasts bundles were taken. As shown in Figure 2j-m, the SA sensor fluorescence clearly matches to the red chlorophyll autofluorescence indicating chloroplast localization.

Figure 2: Confocal images of tobacco epidermal cells to visualize the subcellular localization of SA sensor : (a, e, j) Chlorophyll autofluorescence (red), (b, f, k) SA sensor Cyan florescence, (c, g, l) overlay and (d, h, m) brightfield. Row 1 – SA fluorescence was observed in cell periphery indicating cytoplasmic localization. Row 2 – fluorescence of cells plasmolyzed with 0.8 M mannitol, SA sensor fluorescence observed in the apoplastic space formed by

the shrinking protoplast as indicated by the arrows. (i) Zoom in image of a cell from panel f, (*) indicate the apoplastic space. Row 3 – Overlap of red chlorophyll autofluorescence with cyan indicating chloroplast localization of sensor.

The chlorophyll pigment is present in thylakoids inside the chloroplast. Since there is complete overlap of red fluorescence with the SA sensor fluorescence (Reviewer-only Figure 1), it shows that the SA sensor can cross the chloroplast membrane and enter inside it. It is not easy to specifically determine lumen and stroma based on fluorescence but the sensor need to be in stroma in order to reach the thylakoids. As comparison, Zhang J. et al, J. Cell Biol. 218,8 (2019), 2638–2658 show in the Reviewer-only Figure 2 the scenario of green fluorescent protein (GFP) signals surrounding the chloroplast indicative of chloroplast membrane localization. With our present data, it can be inferred that post infiltration, the sensors are observed in the apoplast, cytoplasm and within chloroplast. Hence, they are suitably present to detect any biosynthesis of SA post stress. The manuscript is also revised accordingly in Line 225 as follows.

Line 225: “SA sensor fluorescence was observed in the cell periphery indicating cytoplasmic localization (Figure 2a-d). To test whether SA sensor localizes to apoplast, plasmolysis was performed. SA sensor fluorescence was seen in the apoplastic space formed by the shrinking protoplast (Figure 2e-i). Based on SA fluorescence overlap with the red chlorophyll autofluorescence, the sensor also localizes within the chloroplasts (Figure 2j-m). As SA biosynthesis is mainly localized to chloroplast and cytoplasm, the SA sensor is suitably localized in cells to detect the overall SA production post stress.”

Reviewer-only Figure 1: Zoom in image showing complete overlap of SA sensor signal with chlorophyll indicating SA sensor can cross the chloroplast membrane and internalize.

Reviewer-only Figure 2 (from Zhang J. et al, J. Cell Biol. 218,8 (2019), 2638–2658): Confocal micrographs after transient expression of GFP fusion proteins in *N. benthamiana* leaves with GFP signal surrounding the chloroplasts, indicative of chloroplast membrane localization. Scale bar are 5 µm.

[Reviewer 1 comment 7] In Figure 4, please explain the detailed method to calculate the SA concentration using the calibration curve (intensity fold-changes; %) in Figure 1g. Because 1) the SA sensor is localized to the various sub-cellular compartments in which basal SA concentrations are different and 2) the total amount of the SA sensor infiltrated into the plant tissue is not estimated, I do not think that the authors could precisely calculate the SA concentrations *in planta*.

[Author response]:

As requested by the reviewer, we have now included the detailed method for SA concentration calculation in Supplementary Information and Supplementary Figure 8 using the Pipecolic Acid treatment data as an example. It is important to note that the average fluorescence intensity is calculated over the entire sensor spot area harbouring many cells as shown in the brightfield image of the leaf (Supplementary Figure 8a). After sensor infiltration, the initial fluorescence intensity (I_0) represents the total basal SA levels present in the cells within the spot area prior to stress ($t = 0$ h), as illustrated by the false-color sensor intensity maps of the imaged leaf area (Supplementary Figure 8b). Representative false-color intensity maps are also shown for the same imaged leaf areas at $t = 2$ h, 4 h and 6 h post treatment with pipecolic acid. By integrating the intensity maps obtained over the 6 h time period, we can obtain fluorescence intensity time curves of the imaged leaf areas. We agree with the reviewer that the total amount of SA sensor uptake into the plant tissue is not precisely calculated. While we inject the same amount of SA sensor solution into the leaves at a fixed sensor concentration, the extent by which the sensor solution is taken up by the leaf and spread throughout the leaf tissue could vary from plant to plant. This causes unavoidable variance in the sensor fluorescence intensities among biological replicates. With multiple replicates, the differences in tissue uptake can be averaged out. More importantly, the sensor fluorescence intensity changes ($I - I_0$) in the same population of cells are calculated using I_0 as normalization. Hence, the normalized intensity time curves (Supplementary Figure 8c) will all have an initial intensity of “1”, regardless of the different basal SA levels in the various sub-cellular compartments and the different amount of SA sensor uptake into the plant tissues. We then focus on detecting the normalized fluorescence intensity deviation from “1” post stress, which is indicative of overall amount of SA produced post stress within the sensor spot area. By taking an intensity ratio between the SA sensor and reference sensor which is non-responsive to SA over time (Supplementary Figure 8d), we further account for other sensor fluctuations unrelated to SA signaling *in planta* post stress. The normalized intensity ratios are then averaged across independent biological replicates. Using ImageJ software (Rasband, W.S., ImageJ, U. S. National Institutes of Health, Bethesda, Maryland, USA, <https://imagej.nih.gov/ij/>, 1997-2018.), SA concentration map images (Supplementary Figure 8e) can also be converted from the respective false-color intensity maps using the SA sensor calibration curve at Figure 1g of $\frac{I_0 - I}{I_0} = A \times \frac{[SA]}{[SA] + K_D}$, where $A = 0.40558$ and $K_D = 31.421 \mu\text{M}$. The normalized intensity ratio time curves can also be converted to SA concentration time curves with the same sensor calibration equation (Supplementary Figure 8f). The above method allows us to precisely calculate the change in SA over the 6 h period post pipecolic acid treatment within the population of cells in the imaged area *in planta*. We have added the above detailed explanation as Supplementary information, Line 55.

Line 55: “The detailed method for SA concentration calculation is explained using the pipecolic acid experiment as an example. Firstly, the average fluorescence intensity is integrated over the entire sensor spot area harbouring many cells as shown in the brightfield image of the leaf (Supplementary Figure 8a). After sensor infiltration, the initial fluorescence intensity (I_0) represents the total basal SA levels present in the cells within the spot area prior to stress ($t = 0$ h), as illustrated by the false-color sensor intensity maps of the imaged leaf area (Supplementary Figure 8b). Representative false-color intensity maps are also shown for the same imaged leaf areas at $t = 2$ h, 4 h and 6 h post treatment with pipecolic acid. By integrating the intensity maps obtained over the 6 h time period, we can obtain fluorescence intensity time curves of the imaged leaf areas. The sensor fluorescence intensity changes ($I - I_0$) in the same population of cells are calculated using I_0 as normalization. The normalized

intensity time curves (Supplementary Figure 8c) will have an initial intensity of “1”. We then focus on detecting the normalized fluorescence intensity deviation post stress, which is indicative of overall amount of SA produced post stress within the sensor spot area. By taking an intensity ratio between the SA sensor and reference sensor which is non-responsive to SA over time (Supplementary Figure 8d), we further account for other sensor fluctuations unrelated to SA signaling *in planta* post stress. The normalized intensity ratios are then averaged across independent biological replicates. Using ImageJ software, SA concentration map images (Supplementary Figure 8e) can be converted from the respective false-color intensity maps by applying the SA sensor calibration curve at Figure 1g of $\frac{I_0 - I}{I_0} = A \times \frac{[SA]}{[SA] + K_D}$, where $A = 0.40558$ and $K_D = 31.421 \mu\text{M}$. The normalized intensity ratio time curves can also be converted to SA concentration time curves with the same sensor calibration equation (Supplementary Figure 8f). This method allows us to precisely calculate the change in SA over the 6 h period post pipelicolic acid treatment within the population of cells in the imaged area *in planta* and has been used consistently to calculate SA concentrations in all time course experiments reported in this paper.”

Supplementary Figure 8: Detailed method to calculate the local SA concentration using Pipecolic Acid treatment data as example. (a) Bright-field images of pak choi infiltrated with the reference sensor and SA sensor to the right and left of leaf midvein respectively with sensor areas represented as overlay; (b) False-color images of SA and reference sensor fluorescence before pip treatment (0 h), and after pip treatment (2, 4 and 6 h) showing gradual quenching of SA sensor while reference sensor remains invariant; (c) Time-plot showing the normalized SWNT intensity of reference (red) and SA (blue) sensors as well as the SA/Ref ratio (green) of 1 of the replicates of pak choi undergoing pip treatment. Average intensity of the 1st 20 frames is used for normalization; (d) Normalized SA/Ref ratios obtained for 5 different replicates of pak choi undergoing pip treatment; (e) Change in local SA concentrations of Pak choi plants before pip treatment (0 h), and after pip treatment (2, 4 and 6 h) derived from false-color intensity maps after applying the SA sensor calibration curve at Figure 1g of $\frac{I_0 - I}{I_0} = A \times \frac{[SA]}{[SA] + K_D}$, where $A = 0.40558$ and $K_D = 31.421 \mu\text{M}$; Normalized fluorescence intensity ratios (red) and the corresponding change in the local SA concentration (blue) measured upon 1 mM pipecolic acid (pip) treatment at $t = 15$ min (black arrow). Shaded regions represent standard error across three independent replicates.

[Reviewer 1 comment 8] In Figure 4c and 4d and Line 263-269, the local SA concentration gradually increased from 0 μM to 6.6 μM (0.95 $\mu\text{g/g}$ FW) at 6 h. I was puzzled by these data. 1) The authors showed that the minimum detection limit of the SA sensor is 4.4 μM (Line 175-177). 2) Given that 4.9 $\mu\text{g/g}$ FW for 10 μM estradiol (Line 224) corresponds to a 0.1-fold quenching change (estimated by Fig 3e), 0.95 $\mu\text{g/g}$ FW should correspond to a 0.02-fold change at the maximum (6 h). I am not convinced that this SA sensor could detect such small amounts of SA (lower than the detection limit; or equal to the noise levels). Indeed, the fluorescent (false color) images are not distinguishable (Figure 4g).

[Author response]:

The concentration of SA determined by nanosensor fluorescence intensity changes cannot be directly correlated with the SA concentrations as determined by LCMS. The SA concentration obtained by nanosensor in Figure 4c corresponds to the local average SA concentration in the population of cells within in the imaged area. For determining SA concentration by LC-MS analysis, the entire leaf was taken for extraction. Hence LC-MS derived concentration should be an overall average of SA produced by individual cells in the entire leaf upon stress whereas the sensor depicts overall average of SA produced by a subpopulation of cells within the leaf. LC-MS was performed to validate the increase in SA production in leaves after stress treatment as indicated by the sensor. Note that this is a clear analytical advantage afforded by the *in planta* nanosensor approach.

The nanosensor experimental design and sensor data processing for Figure 3 and Figure 4 of the previous manuscript are also different and cannot be compared directly. Figure 3 compares the SA levels in mutant Arabidopsis plants subjected to different concentrations of estradiol treatment at the 16 h time-point compared to wild-type Arabidopsis. At the 16 h time-point, a snapshot of the SA sensor fluorescence was measured. While we inject the same amount of SA sensor solution into the leaves at a fixed sensor concentration, the extent by which the sensor solution is taken up by the leaf and spread throughout the leaf tissue could vary from plant to plant. This causes unavoidable variance in the sensor fluorescence intensities among biological replicates. With multiple replicates, we have shown that the variance can be averaged out and Figure 3e shows that the SA sensor exhibits increased quenching with increase in estradiol concentrations, while reference sensor fluorescence remains largely invariant. However, due to the variance, we noted in Figure 3f that the increase is not significant for 10 μM estradiol treatment and wild-type but significant for 50 and 100 μM estradiol treatment. On the other hand, Figure 4 is a dynamic time-course study for 4 h post stress within the same biological replicate with normalization of initial sensor fluorescence intensity prior to stress. This normalization done in Figure 4 effectively reduces the variance and allows us to detect smaller amounts of *in planta* SA produced. However, the normalization cannot be applied to Figure 3, as it is not a time-course study. These have been clarified in the revised manuscript main text, Line 259 and Line 285.

Line 259: “From the false-color images obtained at the 16 h time-point, the SA sensor nIR fluorescence appears progressively dimmer as E2 concentrations increase, while the reference sensor fluorescence remains relatively unchanged (Figure 3e).

Line 285: “The sensor intensities obtained over time are normalized by the initial intensities prior to Xcc infection. The SA sensor fluorescence begins to quench approximately 1 h post infection, indicative of increasing SA accumulation whereas the reference sensor fluorescence remains relatively invariant. The magnitude of fluorescence quenching could be converted to local SA concentrations changes using the calibration curve shown in Figure 1g.

With regards to sensor detection limit *in planta*, although the final total SA concentration of 6.6 μM present in Figure 4c is close to sensor limit of detection of 4.4 μM , it is important to note that the

intensity change across the entire sensor spot area is not equal. Certain regions within in the imaged area show high fluorescence intensity changes while other regions show none, due to the different spatial/ temporal rates of SA production in individual cells upon stress perception. From the SA concentration map images in Figure 4b, it is clear that some regions of the sensor spot area have negligible intensity changes over 6 h while others accumulate higher levels of SA beyond 25 μM , so upon averaging, the final value drops close to limit of detection. The same is observable in the revised Figure 4g for plants that experienced pipelicolic acid treatment. This point has been clarified in the main text, Line 292.

Line 292: “The SA concentration map illustrates that certain regions of the sensor spot area has accumulated higher levels of SA while other regions has negligible SA. At $t = 6$ h, the local SA concentration averaged across the entire sensor spot for 3 independent plant replicates reaches 6.6 μM (Figure 4c).”

We thank the reviewer for pointing out that Figure 4g images showing false-color intensity maps are not immediately distinguishable. We have now re-processed the images and converted them from sensor intensities to local SA concentrations. The spatial production of SA upon pipelicolic acid treatment is now much clearer. The manuscript main text has also been revised accordingly in Line 321.

Line 321: “This fluorescence quenching intensity was converted to the local SA concentration maps shown in Figure 4g. At 6 h, the local SA concentration, detected by averaging across the entire SA sensor spot, reaches 5.0 μM (Figure 4h).”

Figure 4: (f) Bright-field images of pak choi infiltrated with the reference sensor and SA sensor to the right and left of leaf midvein respectively with sensor areas represented as overlay; (g) Change in local SA concentrations of pak choi plants before pip treatment (0 h), and after pip treatment (2, 4 and 6 h); (h) Change in the local SA concentration measured upon 1 mM pip treatment at $t = 15$ min (black arrow). Shaded regions represent standard error across three independent replicates.

[Reviewer 1 comment 9] Line 306 and the title, the authors suggest that “distinct ROS and SA waves are observed.” However, I cannot follow where the ROS and SA waves occurred in Figure 5 and S7. Please provide their spatial and temporal information including velocities of the ROS and SA waves. Convincing pictures and movies (1 frame per 30 s) would help understand the authors’ claim.

[Author response]:

In response to the reviewer’s concern, we have significantly revised Figure 5 of the manuscript and converted the false-color sensor intensity images to time-dependent H₂O₂ and SA concentration map images using the respective nanosensor calibration curves. The manuscript main text has modified in Line 338. To highlight the distinct primary ROS waves induced by each stress, the H₂O₂ concentration maps are shown at 5-min time-points for the first 45 min post stress. Subsequent, H₂O₂ concentration maps are shown at 1 h, 1.5 h/2 h and 4 h time-points to highlight the dynamics of secondary ROS waves induced by heat and light stress. To highlight the distinct onset of SA production induced by each stress, the SA concentration maps are shown at 15-min time intervals for the first 2 h post stress where onset of SA production occurs with the exception of mechanical wounding, followed by 30-min time intervals for the next 2 h. From these new images, the generation of distinct ROS waves and SA can be clearly observed. Supplementary videos of H₂O₂ and SA concentration maps (Supp. Video 1-4) for each type of plant stress treatment are also added in this revised manuscript submission.

Line 338: “Derived H₂O₂ and SA concentration maps obtained for the multiplexed nanosensors, as well as their respective brightfield images, are shown after the pak choi leaves were subjected to mechanical wounding (Figure 5a, Supp. Video 1), *Xcc* infection (Figure 5b, Supp. Video 2), high light (Figure 5c, Supp. Video 3) and high heat (Figure 5d, Supp. Video 4) stresses. Distinct pattern of ROS and SA generation was observed for each type of stress. To highlight the distinct primary ROS waves induced by each stress, the H₂O₂ concentration maps are shown at 5-min time-points for the first 45 min post stress. Subsequently, H₂O₂ concentration maps are shown at 1 h, 1.5 h/2 h and 4 h time-points to highlight the dynamics of secondary ROS waves induced by heat and light stress. To highlight the distinct onset of SA production induced by each stress, the SA concentration maps are shown at 15-min time intervals for the first 2 h post stress where onset of SA production occurs with the exception of mechanical wounding, followed by 30-min time intervals for the next 2 h. These concentration maps reveal the coordination between these two pathways during the respective stress responses.

Figure 6: Bright-field and corresponding H₂O₂ (top row) and SA (bottom row) concentration maps of pak choi infiltrated with reference sensor (green), SA sensor (blue), and H₂O₂ sensor (red) under 785 nm laser excitation. Pak choi plants were subjected to (a) mechanical wounding, (b) *Xcc* infection, (c) high light and (d) high heat treatment respectively. Snapshots of H₂O₂ concentration maps are shown at 5-min intervals for the first 45 min post-stress capturing the first ROS wave, followed by 1 h, 1.5 or 2 h, and 4 h time-points capturing the secondary

ROS wave. Snapshots of SA concentration maps are shown at 15-min intervals for the first 2 h post-stress, followed by 30-min intervals for the next 2 h.

As suggested by the reviewer, we have also calculated velocities of the ROS waves using the same approach as reported in Lew, T.T.S. et al, Nature Plants 6 (2020), 404–415, by taking the ratio of the stress application site distance to the sensor and the mean sensor lag time. Sensor lag time corresponds to the time taken from stress application to when the sensor response drifted away from 5 standard deviations of the baseline. As reported in Lew, T.T.S. et al, Nature Plants 6 (2020), 404–415, ROS wave velocities are dependent on type of stress. For wounding, ROS wave velocities of different plant species tested were found to vary from 0.44 to 3.10 cm/min. In this work, we estimated the ROS wave velocities of pak choi plants subjected to various stresses. H₂O₂ wave velocity was found to be highest for high heat treatment at 1.20 cm/min. Similar wave velocities of 0.19 and 0.20 cm/min were obtained for wounding and Xcc infection respectively, while high light treatment has the lowest velocity at 0.08 cm/min. This is shown in Supplementary Figure 12a and described in the revised manuscript Line 357.

Line 357: “ROS wave velocities were calculated using the same approach as reported in previous work⁴⁰ and found to be dependent on type of stress applied (Supp. Fig. S12a). ROS wave velocity was highest for high heat treatment at 1.20 cm/min, while similar ROS wave velocities of 0.19 and 0.20 cm/min were obtained for wounding and Xcc infection respectively. High light treatment has the lowest ROS wave velocity at 0.08 cm/min.”

Supplementary Figure 12a: Comparison of H₂O₂ wave velocities for pak choi plants that have been subjected to wounding, Xcc infection, high light and high heat stresses. Data are mean ± s.e.m., with *n* = 3 biologically independent samples.

The dynamics of the SA wave differ from those of the ROS wave because SA does not follow an autocatalytic production mechanism as the ROS wave does. Instead, generation of SA appears to be prompted by ROS. In other words, the generation or first appearance of SA travels as a wave, but the concentration profile then evolves via a diffusive mechanism. We can hence track the SA source and estimate the wave velocity by taking a ratio of the stress application site distance to the sensor spot, and the mean SA onset time (Figure 6h). Using this approach, the SA wave velocity is calculated to be highest for high light treatment at 0.044 cm/min, followed by high heat treatment at 0.028 cm/min

and lowest for Xcc infection at 0.011 cm/min. The above is included in the revised manuscript, Line 387.

Line 387: “The dynamics of the SA wave differ from those of the ROS wave because SA does not follow an autocatalytic production mechanism as the ROS wave does. Instead, generation of SA appears to be prompted by ROS. While the generation or first appearance of SA travels as a wave, the subsequent SA concentration profile evolves via a diffusive mechanism. We can hence track the SA source and estimate the SA wave velocity by taking a ratio of the stress application site distance to the sensor spot, and the mean SA onset time (Figure 6h). Using this approach, the SA wave velocity is calculated to be highest for high light treatment at 0.044 cm/min, followed by high heat treatment at 0.028 cm/min and lowest for Xcc infection at 0.011 cm/min (Supp. Fig. S12b).”

Supplementary Figure 12b: Comparison of SA wave velocities for pak choi plants that have been subjected to Xcc infection, high light and high heat stresses. Data are mean \pm s.e.m., with $n = 3$ biologically independent samples.

[Reviewer 1 comment 10] Professor Ron Mittler’s team reported RBOH-dependent ROS waves upon a variety of environmental stresses including light stress. To support the authors’ model, it would be important to test Arabidopsis mutants such as rbohD/f mutants. Once again, one of the big advantages of this new nanosensor is that the authors can promptly conduct the experiments without making transgenic lines.

[Author response]:

We agree with the reviewer of the importance of testing the H₂O₂ nanosensor waves in *Arabidopsis* mutants. The H₂O₂ nanosensor has been previously reported and tested in Arabidopsis mutants (Lew, T.T.S. et al, Nature Plants 6 (2020), 404–415). In that paper, the H₂O₂ nanosensor was used to image wound induced H₂O₂ waves in both wild-type and Arabidopsis mutants including rbohD mutant. Before wounding, no significant difference in the nanosensor fluorescence intensity was detected between wild-type Arabidopsis and the knockout mutants of rbohD. After wounding, wild-type Arabidopsis showed a decrease in fluorescence intensity of the H₂O₂ sensor while the mutant displayed negligible sensor response.

[Reviewer 1 comment 11] Appropriate statistical analyses are needed such as regarding the specificity of the SA sensor (Figure 2b) and dose-dependency (Figure 3) to strengthen the conclusions.

[Author response]:

As suggested by the reviewer, statistical analyses are now added to Figure 1j (previously Figure 2b), as well as all the bar graphs in Figure 3, using one-way ANOVA with a Tukey's HSD test at $p < 0.05$. In addition, statistical analyses for LCMS data is also included in Figure 4d and 4i, using two-tailed unpaired t-test, **** $P < 0.0001$ and *** $P < 0.001$.

[Reviewer 1 comment 12] Line 375, the authors conclude that "This model (Figure 5c-f) effectively captures the key temporal features of the H₂O₂ and SA waves". However, the model predicts single-peak, transient H₂O₂ dynamics, which is not consistent with the bimodal (two or more-peaks or oscillating) H₂O₂ dynamics observed in high light and heat stress responses (Figures 5e and 5f). The authors need to comment on this.

[Author response]:

The reviewer is correct in pointing out that post stress, there can be secondary / tertiary ROS waves as part of the stress response initiated by the plant. In response to the reviewer's comment, we extended our kinetic modeling in an attempt to capture the secondary and tertiary ROS waves that follow the stimulus. As an initial hypothesis, we have modelled the secondary bursts using the same RBOH dependent mechanism as the initial burst. The revised modelled data (Figure 6c-d) fits well with the observed experimental data for high light and high heat treatments, suggesting the feasibility of this mechanism. However, the precise mechanisms and timings of secondary ROS bursts for different stresses will need further biological investigation to improve the modelling process in future studies. We based this hypothesis on the current literature, summarized as follows. It has been proposed that the ROS produced early (within seconds to minutes) in a cell upon stress is utilized for stress perception and activation of stress signaling network. The ROS produced later is the result of activation of these early signaling defense network towards plant stress acclimatization and defense response (Mittler, R. et al, *Nature Reviews Molecular Cell Biology* 23 (2022), 663-679). The later ROS bursts are known to happen at time points ranging from hours to days post stress (Baxter, A. et al, *Journal of Experimental Botany* 65,5 (2014), 1229-1240). Since our aim was to understand early stress induced signaling network, our initial model was applied to capture only the primary ROS bursts and not the later secondary ROS bursts. Majority of the biphasic ROS burst reported and characterized are in response to biotic stress, distinct ROS peaks in response to abiotic stress are not well described (Castro, B. et al, *Nature Plants* 7 (2021), 403-412). Upon biotic stress the initial ROS bursts is reported to happen rapidly within a few minutes while the second massive burst occurs hours later and plays an important role in cell death (Yuan, M. et al, *Nature* 592 (2021), 105-109; Castro, B. et al, *Nature Plants* 7 (2021), 403-412; Mur, L. A. J. et al, *Journal of Experimental Botany* 59,3 (2008), 501-520).

In our study the second ROS bursts upon biotic stress was not observed within the 4 h time period but probably occurs later. The molecular mechanism involved in the generation the secondary ROS are not well elucidated. Few studies have shown the involvement of RBOH in biphasic ROS production during biotic stress (Adachi, H. et al, *Plant Cell* 27,9 (2015), 2645-2663; Wang, Y. et al, *Journal of Proteomics* 251 (2022), 104423). A recent study also suggests the involvement of other factors such as pattern-triggered immune responses contributing to delayed RBOH dependent secondary ROS production (Arnaud, D. et al, *Plant Physiology* 191,4 (2023), 2551-2569). The above is explained in the revised manuscript, Line 464.

Line 464: "Besides the primary ROS burst produced early within minutes upon stress perception, later ROS bursts are also reported due to activation of early defense signaling network⁸. These

secondary/tertiary ROS bursts are known to happen at time points ranging from hours to days post stress⁸⁶. In this study, mild secondary ROS bursts were observed at 2 and 1.5 h post high light and high heat stresses respectively, while no secondary ROS bursts were observed post biotic stress within the 4 h time period. Majority of biphasic ROS bursts reported and characterized are in response to biotic stress, and are not well described in response to abiotic stresses⁸⁴. Upon biotic stress the initial ROS bursts is reported to happen rapidly within a few minutes while the second massive burst occurs hours later and plays an important role in cell death^{84,87,88}. The secondary ROS bursts upon biotic stress in our study might probably occur after the 4 h time period. The molecular mechanism involved in the generation the secondary ROS are not well elucidated. Few studies have shown the involvement of RBOH in biphasic ROS production during biotic stress^{89,90}. A recent study also suggests the involvement of other factors contributing to secondary ROS production⁹¹. As an initial hypothesis, we have modelled the secondary ROS bursts using the same RBOH dependent mechanism as the initial burst. The modelled data (Figure 6c-d) fits well with the observed experimental data for high light and high heat treatments, suggesting the feasibility of this mechanism. However, the precise molecular mechanisms and timings of secondary ROS bursts for different stresses will need further biological investigation to improve the secondary ROS burst modelling process in future studies.

Figure 6: Time-plots of H₂O₂ concentration derived from nanosensor experiments (blue circles) and mathematical modelling (blue lines), as well as SA concentration derived from nanosensor experiments (red circles) and mathematical modelling (red lines), for pak choi plants that are (a) mechanically-wounded, (b) Xcc infected, (c) high light treated, and (d) high heat treated. Each type of stress treatment of the pak choi leaves occurs at approximately t = 10 min (black arrow). Shaded regions represent standard error across three independent replicates. Residual plots of [H₂O₂]_{model} - [H₂O₂]_{experimental} (blue) and [SA]_{model} - [SA]_{experimental} (red) is shown underneath each fit for the different plant stresses

[Reviewer 1 comment 13] The SA and H₂O₂ nanosensors and the reference are infiltrated into the same leaf. I was wondering whether these sensors are diffused/transported throughout the leaf and mixed in planta. If they are mixed, the experimental design is incoherent. Please explain how the authors separate the sensors in a leaf.

[Author response]:

In response to the reviewer's query, we tagged H₂O₂ nanosensors with Cy3 dye which will make it fluorescent and performed sequential confocal scanning to visualize the two nanosensors. Please note that the intracellular localization pattern of both the sensors are similar. We focused on the chloroplast localization of both sensors as it is easy to decipher. The Cy3- tagged H₂O₂ sensor was infiltrated along with the SA sensor separately into two distinct regions of the pak choi leaf as done for the experiments (Supplementary Figure 10). Additionally, the two sensors were mixed together and then infiltrated into another spot on the leaf. The infiltrated spots are shown in below figure. After 5hrs post infiltration, sequential confocal scanning of the three spots was performed. As shown in figure below, the region infiltrated with SA sensor shows no fluorescence from the Cy3-tagged H₂O₂ and conversely the area infiltrated with Cy3-tagged H₂O₂ shows no fluorescence from the SA sensor. The region where both the sensors were mixed and infiltrated, fluorescence from both the sensors could be observed. The above experiment shows that the infiltrated sensors do not migrate out of their infiltrated regions. Hence, the nanosensor will not mix with one another as the areas chosen for infiltration of the two sensors when multiplexed in the same leaf, are separated by big leaf veins. This is consistent with previous findings in Lew, T.T.S. et al, Nature Plants 6 (2020), 404–41 where it was reported that the ROS and reference sensors do not mix when infiltrated into the left and right sides of the leaf mid-vein. The manuscript is revised accordingly at Line 330, and details are provided in supplementary information.

Line 330: "To investigate if during multiplexing, the sensors can move or stay restricted to their respective infiltrated locations in the leaf, confocal imaging of the SA and Cy3-tagged (GT)₁₅-SWNT ROS sensor was performed post infiltration in Pak choi leaves. As shown in Supp. Fig S10, the fluorescence from SA and ROS sensor was found to remain restricted to their respective infiltrated regions and no mixing was observed."

Supplementary Figure 10: (a) Photograph of pak choi leaf illustrating the respective sensor infiltrated regions ; Confocal images of the leaf regions of pak choi leaf infiltrated with (b-f) SA sensor where no fluorescence from Cy3-tagged (GT)₁₅-SWNT (ROS sensor) is observed , (g-k) Cy3-tagged (GT)₁₅ SWNT (ROS sensor) where no fluorescence from SA sensor is observed and (l-p) mixture of SA sensor and Cy3-tagged (GT)₁₅-SWNT (ROS sensor) where fluorescence from both SWNTs are observed

After we re-processed the H₂O₂ and SA concentration map images, it has also become very clear that the sensor fluorescence changes due to changes in ROS and SA levels are restricted to their respective infiltrated regions as both regions are imaged simultaneously. If the sensors had been mixing, we would have observed changes in sensor fluorescence arising from ROS sensor from the SA-sensor infiltrated leaf area for wounding and infection stress. However, as shown in Figure 6a and 6b, the SA sensor showed no fluorescence changes at the early time-points (<1 h) and remains blank while the ROS sensor region showed significant changes in fluorescence. The manuscript is amended to include this explanation in Line 398.

Line 398: “The fact that we can clearly observe separate timing for SA generation for each type of stress further confirms that the multiplexed nanosensors do not diffuse. In particular, for *Xcc* infection, no fluorescence changes were observed in the SA nanosensor area while the ROS wave is on-going.”

[Reviewer 1 comment 14] Line 260, Figure 1f -> Figure 1g?

[Author response]:

We have now mended the typo error from Figure 1f to 1g.

Reviewer #2 (Remarks to the Author):

Use of nanosensors to detect stress signals to enable early stress detection in plants is of importance for nano-enabled agriculture. Here, the manuscript entitled “Decoding early stress signaling waves in living plants using nanosensor multiplexing” tried to develop and use multiplexing nanosensors to detect H₂O₂ and SA signals in stressed plants simultaneously. Although authors claimed these are multiplexing nanosensors. However, the model build in this study is not well described. Also, the verification and validation of the hypothesized model need more efforts. I would recommend authors to do a substantial revision on the manuscript.

[Author response]:

We thank the reviewer for the positive affirmation that our nanosensor work is of importance to nano-enabled agriculture. The multiplexing of H₂O₂ and SA nanosensors will pave the way forward for multiplexing of many nanosensors for simultaneous detection of various plant hormones, aiding in early plant stress detection. In this revised manuscript, we have made significant improvements to our hypothesized model, which provides a better fit to our experimental data. The manuscript has also been substantially revised, along with additional experiments being conducted, to address the concerns raised by the reviewer.

[Reviewer 2 comment 1] Figure 2c, it seems that this is not a good layer of chloroplasts from the leaves. Also, for Figure 2e, it seems that this is a combination confocal image including bright field.

[Author response]:

In response to the reviewer’s comment, we have now obtained improved and higher resolution confocal images (Figure 2) of the SA sensor localization in the chloroplast (Figure 2j-m). Additionally, plasmolysis assay was also conducted to confirm the apoplastic localization of the SA nanosensor (Figure 2e-i). The manuscript under is revised accordingly in Line 225 as follows.

Line 225: “SA sensor fluorescence was observed in the cell periphery indicating cytoplasmic localization (Figure 2a-d). To test whether SA sensor localizes to apoplast, plasmolysis was performed. SA sensor fluorescence was seen in the apoplastic space formed by the shrinking protoplast (Figure 2e-i). Based on SA fluorescence overlap with the red chlorophyll autofluorescence, the sensor also localizes within the chloroplasts (Figure 2j-m). As SA biosynthesis is mainly localized to chloroplast and cytoplasm, the SA sensor is suitably localized in cells to detect the overall SA production post stress.

Figure 2: Confocal images of tobacco epidermal cells to visualize the subcellular localization of SA sensor : (a, e, j) Chlorophyll autofluorescence (red), (b, f, k) SA sensor Cyan florescence, (c, g, l) overlay and (d, h, m) brightfield. Row 1 – SA fluorescence was observed in cell periphery indicating cytoplasmic localization. Row 2 – fluorescence of cells plasmolyzed with 0.8 M mannitol, SA sensor fluorescence observed in the apoplastic space formed by the shrinking protoplast as indicated by the arrows. (i) Zoom in image of a cell from panel f, (*) indicate the apoplastic space. Row 3 – Overlap of red chlorophyll autofluorescence with cyan indicating chloroplast localization of sensor.

[Reviewer 2 comment 2] The compatibility of the used nanosensors in Arabidopsis is not shown.

[Author response]:

As requested by the reviewer, we have investigated the compatibility of the SA sensor in both Arabidopsis and pak choi plants. As shown in Supp. Fig S6, when compared to control water-infiltrated plants, the SA sensor-infiltrated plants displayed no difference in overall growth, no signs of premature senescence and no changes were observed in the chlorophyll content of the leaves over a period of 4 weeks. We have added this information into the revised manuscript, Line 268 and Line 279.

Line 268: “We also investigated the biocompatibility of the SA sensor in *A. thaliana*. As shown in Supp. Fig. S6, when compared to control water infiltrated plants, the SA sensor infiltrated plants showed no changes in the chlorophyll content of the leaves, no signs of premature senescence and no difference in the overall growth of the plant.”

Line 279: “Similar to *A. thaliana*, we examined the biocompatibility of the SA sensor in pak choi plants and found no adverse effects on leaf life span, chlorophyll content and overall plant growth (Supp. Fig. S6).”

Supplementary Figure 6: SA SWNT compatibility evaluation. The SA sensor-infiltrated (a) Arabidopsis and (b) Pak choi plants displayed no difference in overall growth and showed no visible signs of premature senescence compared to control plants; (c,d) chlorophyll content of the SA sensor-infiltrated leaves over a period of 4 weeks is unchanged compared to control plants. (Blue arrows indicate the infiltrated leaf). Scale bar = 3 cm.

The H₂O₂ nanosensor has been previously published in Lew, T.T.S. et al, Nature Plants 6 (2020), 404–415). In that work, similar studies were performed to investigate the compatibility of the ROS nanosensors in plants. After ROS sensor infiltration, no changes were observed in the chlorophyll content of the leaves over a period of 4 weeks and no changes to the life span of the leaves was observed.

[Reviewer 2 comment 3] Different nanosensors are injected into the leaf. Whether the nanosensors can move to other leaf areas which might cause confounding effect?

[Author response]:

In response to the reviewer's query, we tagged H₂O₂ nanosensors with Cy3 dye which will make it fluorescent and performed sequential confocal scanning to visualize the two nanosensors. Please note that the intracellular localization pattern of both the sensors are similar. We focused on the chloroplast localization of both sensors as it is easy to decipher. The Cy3- tagged H₂O₂ sensor was infiltrated along with the SA sensor separately into two distinct regions of the pak choi leaf as done for the experiments (Supplementary Figure 10). Additionally, the two sensors were mixed together and then infiltrated into another spot on the leaf. The infiltrated spots are shown in below figure. After 5hrs post infiltration, sequential confocal scanning of the three spots was performed. As shown in figure below, the region infiltrated with SA sensor shows no fluorescence from the Cy3-tagged H₂O₂ and conversely the area infiltrated with Cy3-tagged H₂O₂ shows no fluorescence from the SA sensor. The region where both the sensors were mixed and infiltrated, fluorescence from both the sensors could be observed. The above experiment shows that the infiltrated sensors do not migrate out of their infiltrated regions. Hence, the nanosensor will not mix with one another as the areas chosen for infiltration of the two sensors when multiplexed in the same leaf, are separated by big leaf veins. This is consistent with previous findings in Lew, T.T.S. et al, Nature Plants 6 (2020), 404–41 where it was reported that the ROS and reference sensors do not mix when infiltrated into the left and right sides of the leaf mid-vein. The manuscript is revised accordingly at Line 330 and details are provided in supplementary information

Line 330: "To investigate if during multiplexing, the sensors can move or stay restricted to their respective infiltrated locations in the leaf, confocal imaging of the SA and Cy3-tagged (GT)₁₅-SWNT ROS sensor was performed post infiltration in Pak choi leaves. As shown in Supp. Fig S10, the fluorescence from SA and ROS sensor was found to remain restricted to their respective infiltrated regions and no mixing was observed."

Supplementary Figure 10: (a) Photograph of pak choi leaf illustrating the respective sensor infiltrated regions ; Confocal images of the leaf regions of pak choi leaf infiltrated with (b-f) SA sensor where no fluorescence from Cy3-tagged (GT)₁₅-SWNT (ROS sensor) is observed , (g-k) Cy3-tagged (GT)₁₅ SWNT (ROS sensor) where no fluorescence from SA sensor is observed and (l-p) mixture of SA sensor and Cy3-tagged (GT)₁₅-SWNT (ROS sensor) where fluorescence signal from both SWNTs are observed

After we re-processed the H₂O₂ and SA concentration map images, it has also become very clear that the sensor fluorescence changes due to changes in ROS and SA levels are restricted to their respective infiltrated regions as both regions are imaged simultaneously. If the sensors had been mixing, we would have observed changes in sensor fluorescence arising from ROS sensor from the SA-sensor infiltrated leaf area for wounding and infection stress. However, as shown in Figure 6a and 6b, the SA sensor showed no fluorescence changes at the early time-points (<1 h) and remains blank while the ROS sensor region showed significant changes in fluorescence. The manuscript is amended to include this explanation in Line 398.

Line 398: “The fact that we can clearly observe separate timing for SA generation for each type of stress further confirms that the multiplexed nanosensors do not diffuse. In particular, for *Xcc* infection, no fluorescence changes were observed in the SA nanosensor area while the ROS wave is on-going.”

[Reviewer 2 comment 4] Figure 3d. Do we have the images at different time points?

[Author response]:

In response to the reviewer's query, we have clarified the reason for choosing the 16 h time-point for sensor studies. This was because 16 h is the time required to get the highest gene expression of *ICS1* transcript upon E2 induction. To investigate the induction rate of *ICS1* transcript, transgenic seedlings were treated with 100 μ M of estradiol and *ICS1* gene expression was measured at different time intervals ranging from 30 min to 96 h. The *ICS1* transcript was detectable at 30 minutes post estradiol treatment, increased to highest level at 16 h and then gradually declined in the transgenic plant. This is shown in Figure 3a. The manuscript is amended to include the above experimental result in Line 246.

Line 246: "To investigate the induction rate of *ICS1* transcript, transgenic seedlings were treated with 100 μ M of E2 and *ICS1* gene expression was measured at different time intervals ranging from 30 min to 96 h (Figure 3a). The *ICS1* transcript was detectable at 30 minutes post E2 treatment, increased to highest level at 16 h and then gradually declined in the transgenic plant. Hence, we decided to perform the sensor studies at 16h time point.

Figure 3a: Complementation of *A. thaliana ICS1* mutant plants with *ICS1* gene driven by an inducible XVE expression system to generate transgenic plants with varying amount of SA hormones and *In planta* validation of SA nanosensor using transgenic *A. thaliana* mutant line (*XVE::ICS1*), where SA levels can be activated and controlled by amount of estradiol (E2) inducer treatment. (a) Maximum expression of gene was detected at 16hrs post induction. Bar graphs show the mean values with error bars representing standard error from independent experiments. Dots represent each data point. Different alphabet letters show significant differences using one-way ANOVA with a Tukey's HSD test at $p < 0.05$.

[Reviewer 2 comment 5] Figure 5. H₂O₂ signals are detected in wounding stress, while no detection of SA was found in wounding stress. I am wondering if authors can do the continuous wounding to see if the SA can be detected by the nanosensor or not.

[Author response]:

Jasmonic acid (JA) is the major hormone that is involved in wounding responses. It is well known that wounding induces a rapid production of JA in plants along with ROS bursts. Studies suggest that JA and SA pathways are mutually antagonistic. JA can negatively regulate SA biosynthesis. The results which we see upon wounding where ROS is present but not SA fits with the published literature. However, SA may be induced at later time-points (hours to day) as part of the defense response and adaptation. In *Arabidopsis*, an increase in SA was observed after 6 h post mechanical wounding while JA increase was observed within minutes of wounding (Ogawa T. et al, *Plant Biotechnology* 27, 2 (2010), 205-209; Kimberlin A.N. et al, *Plant Physiol.* 189, 4 (2022), 1925-1942). As we are interested in capturing the signalling pathways activated early on upon stress perception, we have monitored the plants for only up to 4 h.

[Reviewer 2 comment 6] The fitting curve of SA in Figure 5 d and of H₂O₂ in Figure 5e is not good, suggesting that there is room to optimize the model.

[Author response]:

We have optimized the model in response to the reviewer's comment to improve both the fitting curves of SA and H₂O₂. For SA in Figure 6b, we found that increasing the number of intermediates involved in the production of SA from two to three helped us to describe the data more accurately, as shown in revised Figure 6b. Initially we had arbitrarily chosen a small number of intermediates for the model reaction network as represented in below equation:

In the revised manuscript, equation (4) is now rewritten as:

Besides transcriptional activation of SA biosynthesis by the ICS or PAL pathways, SA can also be rapidly re-converted from its conjugated storage forms, such as SAG and SGE depending on plant needs (Maruri-López, I. et al, *Frontiers in Plant Science* 10 (2019), 423). Hence the intermediate steps involved in generating SA can vary for different stresses. We also note that the experimental data dips into an unphysical slightly negative concentration due to measurement noise, which makes the model fit appear worse than it should be. The above is explained in the revised manuscript, Line 502.

Line 502: "The multiple intermediates involved in producing S (pF , F_1 , and F_2 and F_n) are essential for describing the experimental data in Figure 6b describing the *Xcc* infection response, where the onset of S occurs after the decay of A (Figure 6h). This delay in the onset of S relative to A implies the existence of intermediate chemical species prior to the production of S . We have arbitrarily chosen a small number of intermediates for the model reaction network. Besides transcriptional activation of SA biosynthesis by the ICS or PAL pathways, SA can also be rapidly re-converted from its conjugated storage forms, such as SAG and SGE depending on plant needs¹⁰¹. Hence, the intermediate steps involved in generating SA can vary for different stresses."

For H₂O₂ in Figure 6c and 6d, we have optimized the model to include the secondary / tertiary ROS waves generated as part of the stress response initiated by the plant. In response to the reviewer's comment, we extended our kinetic modeling in an attempt to capture the secondary and tertiary ROS waves that follow the stimulus. As an initial hypothesis, we have modelled the secondary bursts using the same RBOH dependent mechanism as the initial burst. The revised modelled data (Figure 6c-d) fits well with the observed experimental data for high light and high heat treatments, suggesting the feasibility of this mechanism. However, the precise mechanisms and timings of secondary ROS bursts for different stresses will need further biological investigation to improve the modelling process in future studies. We based this hypothesis on the current literature, summarized as follows. It has been proposed that the ROS produced early (within seconds to minutes) in a cell upon stress is utilized for stress perception and activation of stress signaling network. The ROS produced later is the result of activation of these early signaling defense network towards plant stress acclimatization and defense response (Mittler, R. et al, *Nature Reviews Molecular Cell Biology* 23 (2022), 663-679). The later ROS bursts are known to happen at time points ranging from hours to days post stress (Baxter, A. et al, *Journal of Experimental Botany* 65,5 (2014), 1229-1240). Since our aim was to understand early stress induced signaling network, our initial model was applied to capture only the primary ROS bursts and not the later secondary ROS bursts. Majority of the biphasic ROS burst reported and characterized are

in response to biotic stress, distinct ROS peaks in response to abiotic stress are not well described (Castro, B. et al, Nature Plants 7 (2021), 403-412). Upon biotic stress the initial ROS bursts is reported to happen rapidly within a few minutes while the second massive burst occurs hours later and plays an important role in cell death (Yuan, M. et al, Nature 592 (2021), 105-109; Castro, B. et al, Nature Plants 7 (2021), 403-412; Mur, L. A. J. et al, Journal of Experimental Botany 59,3 (2008), 501-520).

In our study the second ROS bursts upon biotic stress was not observed within the 4 h time period but probably occurs later. The molecular mechanism involved in the generation the secondary ROS are not well elucidated. Few studies have shown the involvement of RBOH in biphasic ROS production during biotic stress (Adachi, H. et al, Plant Cell 27,9 (2015), 2645-2663; Wang, Y. et al, Journal of Proteomics 251 (2022), 104423). A recent study also suggests the involvement of other factors such as pattern-triggered immune responses contributing to delayed RBOH dependent secondary ROS production (Arnaud, D. et al, Plant Physiology 191,4 (2023), 2551-2569). The above is explained in the revised manuscript, Line 464.

Line 464: “Besides the primary ROS burst produced early within minutes upon stress perception, later ROS bursts are also reported due to activation of early defense signaling network⁸. These secondary/tertiary ROS bursts are known to happen at time points ranging from hours to days post stress⁸⁶. In this study, mild secondary ROS bursts were observed at 2 and 1.5 h post high light and high heat stresses respectively, while no secondary ROS bursts were observed post biotic stress within the 4 h time period. Majority of biphasic ROS bursts reported and characterized are in response to biotic stress, and are not well described in response to abiotic stresses⁸⁴. Upon biotic stress the initial ROS bursts is reported to happen rapidly within a few minutes while the second massive burst occurs hours later and plays an important role in cell death^{84,87,88}. The secondary ROS bursts upon biotic stress in our study might probably occur after the 4 h time period. The molecular mechanism involved in the generation the secondary ROS are not well elucidated. Few studies have shown the involvement of RBOH in biphasic ROS production during biotic stress^{89,90}. A recent study also suggests the involvement of other factors contributing to secondary ROS production⁹¹. As an initial hypothesis, we have modelled the secondary ROS bursts using the same RBOH dependent mechanism as the initial burst. The modelled data (Figure 6c-d) fits well with the observed experimental data for high light and high heat treatments, suggesting the feasibility of this mechanism. However, the precise molecular mechanisms and timings of secondary ROS bursts for different stresses will need further biological investigation to improve the secondary ROS burst modelling process in future studies.

Figure 6: Time-plots of H₂O₂ concentration derived from nanosensor experiments (blue circles) and mathematical modelling (blue lines), as well as SA concentration derived from nanosensor experiments (red

circles) and mathematical modelling (red lines), for pak choi plants that are (a) mechanically-wounded, (b) *Xcc* infected, (c) high light treated, and (d) high heat treated. Each type of stress treatment of the pak choi leaves occurs at approximately $t = 10$ min (black arrow). Shaded regions represent standard error across three independent replicates. Residual plots of $[H_2O_2]_{\text{model}} - [H_2O_2]_{\text{experimental}}$ (blue) and $[SA]_{\text{model}} - [SA]_{\text{experimental}}$ (red) is shown underneath each fit for the different plant stresses

[Reviewer 2 comment 7] Whether the organelle distribution of nanosensors could affect its detection efficiency on H₂O₂ and SA? The production rate of H₂O₂ and SA is varied in different cell compartments.

[Author response]:

We agree with the reviewer that the production rate of H₂O₂ and SA can vary within cell organelles. We have performed localization experiment to make sure sensors are present within the key compartments where SA/H₂O₂ biosynthesis occurs but we cannot calculate the exact amount of the sensor uptake into each organelles, which may affect its detection efficiency. Our sensors do detect the overall increase in SA/H₂O₂ levels in cells as confirmed by the LC-MS analysis but the precise concentration of increase can be slightly affected by the sensor distribution pattern. Studying the production rate of SA/H₂O₂ levels in various sub-cellular compartments could constitute a future study. This has been included in the revised manuscript, Line 580.

Line 580: "Future work could also include the study of plant hormone production rates within various cell organelles."

[Reviewer 2 comment 8] The quality of figures should be improved. The resolution is low.

[Author response]:

Confocal images (Figure 2) and nanosensor concentration maps (Figure 4 and Figure 5) have now been replaced with higher quality and resolution images in this revised version.

Reviewer #1 (Remarks to the Author):

In this revised manuscript, all the comments raised by the reviewers have been carefully addressed and discussed. The additional new experiments and analyses have strengthened the conclusions of this paper, resulting in a more solid and straightforward logical flow. However, there are still some concerns that require the authors' attention.

Major comments:

(1) The authors claimed in the rebuttal that "there is complete overlap of red fluorescence with the SA sensor fluorescence (Reviewer-only Figure 1)". However, punctuated "cyan" (SA-sensor) fluorescence does not entirely overlap with red chlorophyll autofluorescence in Fig 2i. Furthermore, there is no "cyan" fluorescence in the cytosol and apoplasts in Reviewer-only Figure 1 and Fig 2k. Please carefully determine and describe sub-cellular localizations of the SA sensor in plants. Is the SA sensor localized to the cytosol, apoplasts, chloroplasts and unknown punctuated endomembrane compartments in epidermal cells (Fig 2i), while being localized solely in chloroplasts, not in the cytosol and apoplasts, in mesophyll cells (Reviewer-only Figure 1 and Fig 2k)?

(2) Although in vitro screening using a spectrofluorometer coupled with an InGaAs detector and in vivo imaging using an InGaAs detector with a Nikon lens were well described, information on the optical setting for confocal microscopy (Leica, SP8) is missing. How did the authors visualize the SA sensor and autofluorescence at the cellular level? Given that most detectors in confocal microscopes are less sensitive to near-infrared fluorescence, did the authors add a new detector and a 785 nm excitation laser to SP8? Please provide detailed optical information such as lasers and detectors.

(3) The analysis of changes in the sub-cellular SA levels is crucial. Please provide high-resolution images/videos illustrating changes in SA sensor fluorescence at the cellular level. While the authors presented averaged SA levels in the cytosol, apoplasts and chloroplasts presumably with different basal SA levels, readers would like to know which SA levels change in response to environmental stimuli.

Reviewer #2 (Remarks to the Author):

Dear editor,

Authors have addressed most of my comments. The quality of the manuscript is significantly improved. I have few minor comments (see below) for authors. After addressing it properly, I think the manuscript can be accepted by Nature Communications.

Minor comments.

1. Fig. 2 j,k,l,m. Are you sure these images represent tobacco epidermal cells? For me, this is more like a mesophyll cell layer. Please double check it.
2. Figure 5. Authors claimed that the sensors are not interfered by each other. However, if this measurement is not done in 240 min but in 24 hours or even 240 hours, whether the sensors will distributed to other areas? A discussion is needed.
3. In methods section, authors used a stainless-steel metal rod pre-heated at 50 degree C to touch the tip of pak choi leaf for 30s. I am wondering if this treatment can well represent the heat stress in plants. What's the actual leaf temperature after doing this touching experiment? Also, 30 seconds treatment to mimic heat stress and then followed by 240 min measurement. At 240 min, what will be the real leaf temperature? At least, a justification is needed here.
4. Can authors show more cells in Supplementary Fig. 10.

Reviewer #1 (Remarks to the Author):

In this revised manuscript, all the comments raised by the reviewers have been carefully addressed and discussed. The additional new experiments and analyses have strengthened the conclusions of this paper, resulting in a more solid and straightforward logical flow. However, there are still some concerns that require the authors' attention.

[Author response]:

We thank the reviewer for the positive comments on our revisions and revised experiments. We agree with the reviewer that they have improved the paper. We have made further revisions to the manuscript to address the remaining concerns below.

[Reviewer 1 comment 1] The authors claimed in the rebuttal that “there is complete overlap of red fluorescence with the SA sensor fluorescence (Reviewer-only Figure 1)”. However, punctuated “cyan” (SA-sensor) fluorescence does not entirely overlap with red chlorophyll autofluorescence in Fig 2i. Furthermore, there is no “cyan” fluorescence in the cytosol and apoplasts in Reviewer-only Figure 1 and Fig 2k. Please carefully determine and describe sub-cellular localizations of the SA sensor in plants. Is the SA sensor localized to the cytosol, apoplasts, chloroplasts and unknown punctuated endomembrane compartments in epidermal cells (Fig 2i), while being localized solely in chloroplasts, not in the cytosol and apoplasts, in mesophyll cells (Reviewer-only Figure 1 and Fig 2k)?

[Author response]:

We agree with the reviewer that punctuated “cyan” (SA-sensor) fluorescence does not appear to entirely overlap with red chlorophyll autofluorescence in Fig 2i and there is no “cyan” fluorescence in the cytosol and apoplasts in reviewer-only Figure 1 and Figure 2k. This is due to the confocal imaging being performed on fresh unprocessed leaf samples. Epidermal cells, mesophyll cells and cell organelles are at different depths and focal planes and getting many cells and organelles focused on a single image makes the localization pattern less clear. Because of this we decided to focus separately on each type of cells/organelles to clearly depict the subcellular localization in Figure 2. In row 1 of Figure 2, the focus is mainly to capture the epidermal cells showing SA sensor localization to cytoplasm and in row 2, the focus is to capture the apoplast and cytoplasm after plasmolysis. In Figure 2i, which is a zoomed image from Figure 2g we have focused on the cytoplasm and apoplast to show the localization in cytoplasm and apoplast of epidermal cells. Hence many epidermal chloroplasts which are not in correct focus will show weaker fluorescence. In Figure 2i, the autofluorescence from chloroplast is weaker than the cyan fluorescence from SA sensor at that zoomed area. Hence the red autofluorescence denoting the chloroplast does not come through well in the merged image. The punctuated cyan fluorescence, as observed from the brightfield image (Figure 2h), predominantly aligns with the chloroplasts. For clarity, we have marked out the zoomed area in the brightfield image in revised Figure 2h with a dotted cyan box. The big punctuated red shade on the left of Figure 2i is most likely a reflection of the red laser on leaf surface and does not represent a chloroplast. In Figure 2 row 3 and Reviewer-only Figure 1, the focus is solely to capture chloroplast autofluorescence and to confirm chloroplast localization. Hence, the cytoplasm is not clear. Since these cells are not plasmolyzed, the apoplast cannot be visualized as well. To denote chloroplast localization, we decided to focus on mesophyll cells because majority of chloroplasts are found in the leaf mesophyll tissues. Except for guard cells, leaf epidermal cells do not contain a significant number of chloroplasts. Epidermal chloroplasts are also smaller than the chloroplast present in mesophyll cells. For clarity, Figure 2 caption and Line 226 of the manuscript was revised accordingly as follows.

“Figure 2: Confocal images of tobacco leaf infiltrated with SA sensor, to visualize the subcellular localization of SA sensor: (a, e, j) Chlorophyll autofluorescence (red), (b, f, k) SA sensor Cyan fluorescence, (c, g, l) overlay and (d, h, m) brightfield. Row 1 – SA fluorescence was observed in epidermal cell periphery indicating cytoplasmic localization. Row 2 – plasmolyzed epidermal cells (with 0.8 M Mannitol), showing SA sensor fluorescence in the apoplastic space formed by the shrinking protoplast as indicated by the arrows. (i) Zoom-in overlay image of a cell from Figure 2g with corresponding brightfield image area marked with dotted cyan box in Figure 2h, (*) indicate the apoplastic space. Row 3 – overlap of red chlorophyll autofluorescence with cyan SA fluorescence in mesophyll cells indicating chloroplast localization of sensor.”

Line 226: “SA sensor fluorescence was observed in the cell periphery of epidermal cells indicating cytoplasmic localization (Figure 2a-d). To test whether SA sensor localizes to apoplast, plasmolysis was performed. SA sensor fluorescence was seen in the apoplastic space formed by the shrinking protoplast in epidermal cells (Figure 2e-i). Based on SA fluorescence overlap with the red chlorophyll autofluorescence in mesophyll cells, the sensor also localizes within the chloroplasts (Figure 2j-m).”

In Reviewer-only Figure 3 below, we have tried to show both mesophyll cells and epidermal cells along with cytoplasm and apoplast and chloroplast in plasmolyzed cells in one image. In this image, epidermal cells are at the bottom of the figure with wavy margin and fewer chloroplasts. Cytoplasm and apoplast localization of SA sensor in the epidermal cells are indicated by purple and blue arrows respectively. Chloroplast localization can also be observed in epidermal cells. Similar localization pattern can be observed in mesophyll cells. The more circular cells with many chloroplasts at the top part of the figure are mesophyll cells. In mesophyll cells, localization can be seen in chloroplast. The red arrow indicate mesophyll cytoplasm and green arrow mesophyll apoplast. Hence, upon infiltration into the leaf, Reviewer-only Figure 3 shows that the SA sensor is detected in the epidermal apoplast, cytoplasm, and chloroplasts, as well as in the cytoplasm and chloroplasts of mesophyll cells at the same time. Kindly note that upon plasmolysis, the fluorescence signal in apoplasts is expected to

appear weaker due to increased diffusion and dilution of sensor nanoparticles into the expanded apoplast space. We believe that showing localization of sensor as separate images in Figure 2 would provide better clarity compared to showing all in one image in one frame, with less clarity for each cell type and cell organelle.

Reviewer-only Figure 3: Confocal micrograph of tobacco leaf infiltrated with SA sensor showing the overlay of chlorophyll autofluorescence (red) and SA sensor visible fluorescence (cyan) to visualize the chloroplast localization of SA sensor in both epidermal and mesophyll cells. Cytoplasm and apoplast in the epidermal cells are indicated by purple and blue arrows respectively, while cytoplasm and apoplast in the mesophyll cells are indicated by red and green arrows respectively.

Endoplasmic reticulum (ER) and Golgi constitute the major endomembrane cell organelles. Localization pattern of ER marker GFP-HDEL and Golgi marker ST-GFP in tobacco epidermal cells are shown below in Reviewer-only Figure 4. ER has a distinct reticulate localization pattern and Golgi localization appears as dot like pattern. The cyan fluorescence of the SA sensor does not show similar patterns in our confocal micrographs. Since SA synthesis primarily occurs in cytoplasm and chloroplasts, minor associations of SA sensor with other organelles should not significantly impact SA detection upon stress perception in our experiments.

Reviewer-only Figure 4 (from Noronha, H. et al, PLoS ONE 11(8): e0160976): Confocal micrographs after expression of GFP in tobacco epidermal cells with GFP signal localization in Golgi (left) and ER (right)

[Reviewer 1 comment 2] Although in vitro screening using a spectrofluorometer coupled with an InGaAs detector and in vivo imaging using an InGaAs detector with a Nikon lens were well described, information on the optical setting for confocal microscopy (Leica, SP8) is missing. How did the authors visualize the SA sensor and autofluorescence at the cellular level? Given that most detectors in confocal microscopes are less sensitive to near-infrared fluorescence, did the authors add a new detector and a 785 nm excitation laser to SP8? Please provide detailed optical information such as lasers and detectors.

[Author response]:

As suggested by the reviewer, we have now included detailed information of optical settings used for confocal microscopy in the Online methods section, Line 689. The SA sensor was not visualized by tracking its SWNT near-infrared fluorescence. Instead, we have tracked the visible fluorescence of its polymer wrapping for confocal imaging. The SA sensor comprises SWNT that is non-covalently wrapped with S3 polymer, a conjugated polyfluorene co-polymer. In addition to the intrinsic SWNT near-infrared fluorescence, the polymer has a band gap with λ_{\max} (absorption) at 370 nm and λ_{\max} (emission) at 435 nm. Poly-fluorene co-polymers previously reported in Tang, C.G., Ang, M.C.Y. et al, Nature 539 (2016), 536–540 and Ang, M.C.Y. et al, J. Mater. Chem. C 8 (2020), 124-131, also have similar λ_{\max} , attributed to the co-polymer's π - π^* band. Line 222 of the manuscript was also revised accordingly. We have observed that non-covalent wrapping of SWNT with the polymer did not cause significant quenching or wavelength shifts to the polymer's visible fluorescence. Hence, the SA sensor absorbance band ranges from 350 to 420 nm while its emission band ranges from 415 to 465 nm. Note that free polymer was carefully removed prior to infiltration into tobacco leaf, using repeated centrifugal filtration, to ensure that the visible fluorescence detected in the confocal microscopy images solely originates from the polymer-wrapped SWNTs and not from the free polymer.

Line 222: “Due to conjugated polymer SWNT wrapping, the SA sensor has visible fluorescence attributed to the co-polymer's π - π^* band.”

Line 689: “After synthesis of SA sensor, S3 free polymer that is unbound to SWNTs were removed by centrifugal filtration using the Amicon® Ultra-4 Centrifugal Filter tubes with 100 kDa MWCO. Complete removal of free polymer is confirmed by the absence of the polymer UV-Vis absorbance band (λ_{\max} = 370 nm) in the filtrate. The SA sensor solution was then diluted to 1.25 mg/L and infiltrated into *N. benthamiana* / pak choi leaves. Infiltrated plants were kept in dark at 28 °C for 1hr. Plasmolysis of leaf was induced by treatment with 0.8M Mannitol solution (Sigma-Aldrich, Singapore). All leaf samples were viewed with an inverted confocal microscope (SP8, Leica, Germany). For confocal imaging, the excitation laser and detector wavelength ranges used for each fluorophore is as follows, SA sensor: 405nm (415–440 nm); Cy3-(GT)₁₅ SWNT: 552nm (565–580 nm); Chlorophyll autofluorescence: 594 nm (640–700 nm).”

[Reviewer 1 comment 3] The analysis of changes in the sub-cellular SA levels is crucial. Please provide high-resolution images/videos illustrating changes in SA sensor fluorescence at the cellular level. While the authors presented averaged SA levels in the cytosol, apoplasts and chloroplasts presumably with different basal SA levels, readers would like to know which SA levels change in response to environmental stimuli.

[Author response]:

We agree with the reviewer that it will be interesting to observe changes in SA level at the sub-cellular level post-stress. However, we currently lack the NIR confocal imaging system to decipher changes of SA at the sub-cellular level. While we can use the SA sensor polymer's visible fluorescence to track its

localization, the visible fluorescence intensity does not change upon detection of SA. Only the SWNT NIR fluorescence was found to quench when bound to SA. As the reviewer had mentioned in comment 2, detectors in confocal microscopes tend to be less sensitive to NIR fluorescence. Hence, we are only able to monitor changes in overall averaged SA levels in cells post stress, while analysis of changes in sub-cellular SA levels will be part of a future study. This is included in the revised discussion section, Line 584.

Line 584: "Future work with the nanosensors could also include the study of plant hormone production rates within various cell organelles using nIR confocal microscopes, to attain insights into the dynamic changes of various plant hormones within distinct organelles."

Reviewer #2 (Remarks to the Author):

Dear editor, authors have addressed most of my comments. The quality of the manuscript is significantly improved. I have few minor comments (see below) for authors. After addressing it properly, I think the manuscript can be accepted by Nature Communications.

[Author response]:

We thank the reviewer for the kind remarks and the positive acknowledgement of our manuscript improvements. The comments raised by the reviewer have been addressed below.

[Reviewer 2 comment 1] Fig. 2 j,k,l,m. Are you sure these images represent tobacco epidermal cells? For me, this is more like a mesophyll cell layer. Please double check it.

[Author response]:

We agree with the reviewer that Figure 2 j,k,l,m are indeed mesophyll cell layer. Figure 2 caption has now been amended accordingly as follows.

“Figure 2: Confocal images of tobacco leaf infiltrated with SA sensor to visualize the subcellular localization of SA sensor: (a, e, j) Chlorophyll autofluorescence (red), (b, f, k) SA sensor Cyan fluorescence, (c, g, l) overlay and (d, h, m) brightfield. Row 1 – SA fluorescence was observed in epidermal cell periphery indicating cytoplasmic localization. Row 2 – plasmolyzed epidermal cells (with 0.8 M Mannitol), showing SA sensor fluorescence in the apoplastic space formed by the shrinking protoplast as indicated by the arrows. (i) Zoom-in overlay image of a cell from Figure 2g with corresponding brightfield image area marked with dotted cyan box in Figure 2h, (*) indicate the apoplastic space. Row 3 – overlap of red chlorophyll autofluorescence with cyan SA fluorescence in mesophyll cells indicating chloroplast localization of sensor.”

[Reviewer 2 comment 2] Figure 5. Authors claimed that the sensors are not interfered by each other. However, if this measurement is not done in 240 min but in 24 hours or even 240 hours, whether the sensors will distribute to other areas? A discussion is needed.

[Author response]:

The reviewer is correct in pointing out that the interference studies have not been performed for longer time points. As we are investigating early stress responses in terms of ROS and SA production, the sensor interference was also evaluated for a maximum of 5 h post infiltration. This 5 h time-point was chosen as the sensor multiplexing experiments in Figure 5 were completed within 4 h post-stress and sensors were infiltrated 1 h prior to stress application. We agree with the reviewer that it is important to know if sensors will diffuse to other areas of the leaf at longer time periods, especially if the sensor multiplexing experiments are to be conducted over longer durations. Hence, confining the multiple sensors within microneedle arrays will be essential in preventing sensor diffusion and interference over days. This will constitute part of a future expanded study. The above has been added to the revised manuscript in the Discussion section, Line 578.

Line 578: “This study paves the way for multiplexing of many CoPhMoRe nanosensors for simultaneous detection of various plant analytes following an external stimulus with the use of microneedle arrays embedded with different sensors. As we multiplex more sensors and monitor the plant analytes over longer periods of time in vivo, it will be essential to evaluate the life span, stability, as well as diffusion, of nanosensors within plant cells over days. Confinement of the nanosensors within separate microneedles can help prevent sensor diffusion and interference.”

[Reviewer 2 comment 3] In methods section, authors used a stainless-steel metal rod pre-heated at 50 degree C to touch the tip of pak choi leaf for 30s. I am wondering if this treatment can well represent the heat stress in plants. What's the actual leaf temperature after doing this touching

experiment? Also, 30 seconds treatment to mimic heat stress and then followed by 240 min measurement. At 240 min, what will be the real leaf temperature? At least, a justification is needed here.

[Author response]:

In response to the reviewer’s query, we wish to highlight that the heat stress treatment does not mimic natural heat stress conditions. Instead, the heat stress applied is a localized one, similar to other localized heat stress/injury methods reported in Lew, T.T.S. et al, *Nature Plants* 6 (2020), 404–415, Szechyńska-Hebda, M. et al, *The Plant Cell*, 34, 8 (2022), 3047–3065 and Fichman, Y. et al, *PNAS*, 120, 31 (2023), e2305496120. We wanted to keep the heat stress treatment similar to previously published work for better comparison. At 240 min, the real leaf temperature will not have changed from control plants and will probably remain at room temperature. The main objective is to demonstrate the versatility of the nanosensors to detect the unique profiles of ROS and SA generation upon different types of stress treatments in non-model crops. For clarity, we have amended the manuscript text in the Online Methods section, Line 761, to refer to the heat stress as a localized heat stress treatment.

Line 761: “A localized heat stress treatment was conducted by placing the end of a stainless-steel metal rod that is pre-heated in a water bath to 50°C in contact with the tip of the pak choi leaf for 30 s.”

[Reviewer 2 comment 4] Can authors show more cells in Supplementary Fig. 10.

[Author response]:

We have now replaced the figures in Supplementary Fig. 10(b-f) to show more cells to match with the other figures.

Supplementary Figure 10: (a) Photograph of pak choi leaf illustrating the respective sensor infiltrated regions ; Confocal images of the leaf regions of pak choi leaf infiltrated with (b-f) SA sensor where no fluorescence from Cy3-tagged (GT)₁₅-SWNT (ROS sensor) is observed , (g-k) Cy3-tagged (GT)₁₅-SWNT (ROS sensor) where no fluorescence from SA sensor is observed and (l-p) mixture of SA sensor and Cy3-tagged (GT)₁₅-SWNT (ROS sensor) where fluorescence signal from both SWNTs are observed

Reviewer #1 (Remarks to the Author):

The authors have addressed most of the criticisms and conducted several of the suggested experiments. The data presented by the authors are convincing and support their conclusions. I do not have any significant amendments to suggest.